# Hgformer: Hyperbolic Graph Transformer for Collaborative Filtering

**Xin Yang** [1]  **Xingrun Li** [2]  **Heng Chang** [3]  **Jinze Yang** [4]  **Xihong Yang** [5]  **Shengyu Tao** [3]  **Maiko Shigeno** [1]
**Ningkang Chang** [2]  **Junfeng Wang** [5]  **Dawei Yin** [5]  **Erxue Min** [5]

## Abstract

Recommender systems are increasingly spreading to different areas like e-commerce or video streaming to alleviate information overload. One of the most fundamental methods for recommendation is Collaborative Filtering (CF), which leverages historical user-item interactions to infer user preferences. In recent years, Graph Neural Networks (GNNs) have been extensively studied to capture graph structures in CF tasks. Despite this remarkable progress, local structure modeling and embedding distortion still remain two notable limitations in the majority of GNN-based CF methods. Therefore, in this paper, we propose a novel Hyperbolic Graph Transformer architecture, to tackle the long-tail problems in CF tasks. Specifically, the proposed framework is comprised of two essential modules: 1) Local Hyperbolic Graph Convolutional Network (LHGCN), which performs graph convolution entirely in the hyperbolic manifold and captures the local structure of each node; 2) Hyperbolic Transformer, which is comprised of hyperbolic cross-attention mechanisms to capture global information. Furthermore, to enable its feasibility on large-scale data, we introduce an unbiased approximation of the cross-attention for linear computational complexity, with a theoretical guarantee in approximation errors. Empirical experiments demonstrate that our proposed model outperforms the leading collaborative filtering methods and significantly mitigates the long-tail issue in CF tasks. Our implementations are available in https://github.com/EnkiXin/Hgformer.

[1]University of Tsukuba [2]Kyoto University [3]Tsinghua University [4]University of Tokyo [5]Baidu Inc.. Correspondence to: Erxue Min <erxue.min@gmail.com>.

*Proceedings of the $42^{nd}$ International Conference on Machine Learning*, Vancouver, Canada. PMLR 267, 2025. Copyright 2025 by the author(s).

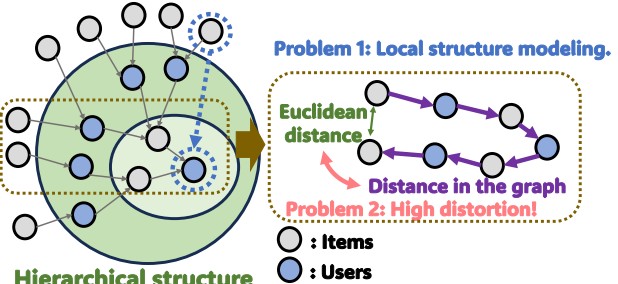

*Figure 1.* This figure highlights two key challenges that traditional GNNs often face when dealing with long-tail items. The first one is the message-passing paradigm in local neighborhoods, which prevents the model from transferring information from less popular items to most users, leading to poor recommendations for these items. The second one is that the embeddings in Euclidean space may fail to capture the long distance in graphs, and will lead to serious information distortion in graph structure.

## 1. Introduction

Recommender systems have become an indispensable part of our daily life, serving as fundamental tools for personalized information filtering and prioritization (Cheng et al., 2016; Davidson et al., 2010; Gong et al., 2020; Min et al., 2025; Xihong et al., 2025; Yang et al., 2025). The core of a recommender system is to predict whether a user will engage with an item, such as by clicking, rating, or purchasing it. In this context, Collaborative Filtering (CF) (Rendle et al., 2012; Xue et al., 2017; Zheng et al., 2018; He et al., 2020), which leverages past interactions between users and items to make these predictions, remains an essential component to deliver effective personalized recommendations. The interaction patterns between users and items in CF tasks naturally form a graph structure, motivating researchers to investigate the use of Graph Neural Networks (GNNs) (He et al., 2020; Wang et al., 2019a; Sun et al., 2021), which has proven significant advantages in modeling graph structures(Kipf & Welling, 2016; Hamilton et al., 2017).

Despite the significant attention and fruitful outcomes in this field, most existing GNNs normally assume that the degree of each node is balanced. However, data in the realm of recommender systems generally exhibit a long-tail distribution (Park & Tuzhilin, 2008; Park, 2012; Yin et al., 2012; Zhao et al., 2023): a small portion of items are highly popular with numerous users, whereas most other items

attract relatively few users. Recent studies have also shown that GNNs-based methods perform well in recommending popular items (head items), but often struggle to perform as effectively with less popular items (tail items) (Sun et al., 2021; Yang et al., 2022a). This issue primarily arises from two factors, which are described in Fig. 1: **i) Local structure modeling:** GNN-based models normally follow the neighborhood aggregation scheme and tend to be biased towards nodes with high degree (Liu et al., 2023). More precisely, in the task of CF, head items that are interacted with by many users typically have higher-quality representations, while the vast majority of tail nodes with few interactions are likely to be underrepresented (Yun et al., 2022). Several GNN frameworks have been proposed to mitigate these degree biases by introducing designated architectures or training strategies specifically for low-degree nodes (Sun et al., 2021; Yun et al., 2022; Zhao et al., 2023), but they still fail to capture the global information as transformers (Ying et al., 2021; Wu et al., 2022; 2023; 2024) and thus achieve sub-optimal results. **ii) Embedding distortion:** Most existing methods encode items and users into Euclidean space (Wang et al., 2019a; He et al., 2020), which is a flat geometry with a polynomial expanding capacity. Data with the long-tail distribution can be traced back to hierarchical structures (Ravasz & Barabási, 2003), whose number of neighbors increases exponentially. As a result, encoding these data via Euclidean space naturally incurs information loss and could subsequently deteriorate the performance of downstream tasks. In contrast, hyperbolic manifold, a non-Euclidean space characterized by constant negative curvature, allows the space to expand exponentially with the radius, making it particularly well-suited for representing tree-like or hierarchical structures (Chami et al., 2019; Ganea et al., 2018). For this reason, in recent years, significant advances have been made in hyperbolic neural networks to better handle the problem of long-tail distribution (Chami et al., 2019; Zhang & Wu, 2023; Yue et al., 2023).

So far, most research in recommender systems has focused on addressing either one of the two problems but has not managed to tackle both simultaneously. Inspired by the successful application of Graph Transformers in graph and node classification tasks (Ying et al., 2021; Wu et al., 2022; 2024; Yang et al., 2024a), this study proposes a novel Graph Transformer architecture in the hyperbolic manifold for Collaborative Filtering. Although the idea of extending Graph Transformers to hyperbolic space for recommendations is intriguing, it poses several challenges that must be overcome:

● **No well-defined parameter-free graph convolution in hyperbolic space.** Parameter-free message-passing paradigms such as LightGCN (He et al., 2020) have shown superior advantages in CF, however, most existing hyperbolic variants of LightGCN, such as HGCF, HRCF, and HICF (Sun et al., 2021; Yang et al., 2022b;a) require to first project embeddings in the hyperbolic manifold back to the tangent space for subsequent graph convolution, which causes information loss and limits their performance.

● **No well-defined hyperbolic self-attention mechanism for collaborative filtering.** Although there are currently some definitions of hyperbolic attention (Zhang et al., 2021a; Yang et al., 2024a), in the field of Collaborative Filtering, the user-item interaction structure is represented as a bipartite graph, existing methodologies are inadequate, and hyperbolic self-attention mechanism tailored for CF tasks has not yet been investigated.

● **Scalability issue of hyperbolic self-attention mechanism.** In recommender systems, real-world graphs are often large-scale, which poses a significant challenge in efficiency when applying transformer architectures with quadratic time complexity. Although many studies have tackled the scalability problem of graph transformers in Euclidean space (Wu et al., 2022; 2023; 2024), the solution for linear computational complexity of hyperbolic self-attention is still under-explored.

To solve these challenges, we propose a new **H**yperbolic **G**raph Trans**former** framework called **Hgformer**. For the first challenge, we propose the Light Hyperbolic Graph Convolutional Network (LHGCN), which performs graph convolution entirely in the hyperbolic manifold. For the second challenge, we propose a novel hyperbolic transformer architecture tailored for CF tasks, which consists of a cross-attention layer and a hyperbolic normalization layer; For the last challenge, we propose an unbiased approximation approach to reduce the computational complexity of hyperbolic self-attention to the linear level. Numerical experiments show that our proposed model performs better than the leading CF models and remarkably mitigates the long-tail issue in CF tasks. We summarize our contributions as follows:

- We propose a novel graph convolution method, LHGCN, which removes the feature transformation and non-linear activation functions from HGCNs and ensures that the entire message passing can be performed fully in hyperbolic manifold without mapping back to Euclidean space.

- We propose a hyperbolic graph transformer framework for CF tasks, which can simultaneously address the challenges of local structure modeling and embedding distortion.

- We propose a kernel function approximation method in hyperbolic space, which reduces the computational complexity of hyperbolic self-attention to a linear scale and we theoretically prove that the approximation error remains within a controllable range.

• The numerical experiments show our proposed method is superior to leading CF models. Moreover, compared with traditional hyperbolic graph neural network methods, our approach can further enhance the model's performance on long-tail items.

## 2. Methods

In this section, we will elaborate on our proposed method. All the notations of this paper are summerized in Appendix 2. We formally define our tasks as follows: **Input:** The interaction graph of users and items $\mathcal{G} = (\mathcal{V}_u, \mathcal{V}_i, \mathcal{E}), \mathcal{E} \subseteq \mathcal{V}_u \times \mathcal{V}_i$. **Output:** A learned function $\mathcal{F} = (u, i | \mathcal{G}, \Theta)$, where $u \in V_u, i \in V_i$ and $\Theta$ denote the model parameters.

As is shown in Fig. 2(a), in our framework, we first map the users and items into embedding space according to their IDs and use an exponential map to project the embeddings into hyperbolic space. To capture the local structure of nodes in the user-item interaction graph, we design a Light Hyperbolic Graph Convolutional Network (LHGCN) and to capture the global structure of the entire interaction graph, we propose a novel hyperbolic transformer, which is composed of a hyperbolic cross-attention mechanism with linear computation complexity and a hyperbolic normalization layer(Bdeir et al., 2023). In the final step, we aggregate the local structure information and the global information from both perspectives for prediction, optimized by a hyperbolic margin-ranking loss.

### 2.1. Hyperbolic Embedding

We first encode each user and item into the embedding space, which is denoted as $\mathbf{u}^{\mathcal{E}} = [\mathbf{u}_1^{\mathcal{E}}; \ldots; \mathbf{u}_N^{\mathcal{E}}]$ and $\mathbf{i} = [\mathbf{i}_1^{\mathcal{E}}; \ldots; \mathbf{i}_M^{\mathcal{E}}]$, where the superscript $\mathcal{E}$ means Euclidean space, $N$ and $M$ means the number of users and items. Then, we use an exponential map to project the embeddings into the hyperbolic space:

$$\mathbf{u}_k^{\mathcal{H}} = \text{Exp}_o^K((0, \mathbf{u}_k^{\mathcal{E}})), \quad \mathbf{i}_k^{\mathcal{H}} = \text{Exp}_o^K((0, \mathbf{i}_k^{\mathcal{E}})).$$

For the detailed definition of the exponential map, see Definition B.6 in Appendix B. Since in the subsequent sections, we will only use the embeddings within the hyperbolic manifold, for convenience and to avoid confusion, we ignore the superscript $\mathcal{H}$ and simply denote the embeddings of users and items in the hyperbolic manifold as $\mathbf{u} = [\mathbf{u}_1; \ldots; \mathbf{u}_N]$ and $\mathbf{i} = [\mathbf{i}_1; \ldots; \mathbf{i}_M]$.

### 2.2. LHGCN: Light Hyperbolic Graph Convolutional Networks

In CF tasks, each node (user or item) is represented by a unique ID, which lacks concrete semantics beyond being an identifier. In such scenarios, LightGCN (He et al., 2020) empirically demonstrated that performing multiple layers of nonlinear feature transformation does not provide ben-

efits and increases the difficulty of model training. Therefore, removing nonlinear feature transformation from GCNs is a well-accepted approach in CF. To perform message-passing in hyperbolic space, a straightforward solution is to first project embeddings in the hyperbolic manifold back to the tangent space at the north pole point, performing parameter-free graph convolution, and then map them back to the hyperbolic manifold (Sun et al., 2021; Yang et al., 2022b;a). However, since the tangent space at the north pole point is merely a local approximation of the north pole point (Boumal, 2023) this can cause a certain degree of information loss. For this sake, we design a simple yet efficient graph convolution method tailed for CF called LHGCN. Similar to HGCF (Sun et al., 2021) and LightGCN (He et al., 2020), LHGCN does not have trainable parameters and all computations are performed entirely on the hyperbolic manifold, eliminating the need for transformations between the hyperbolic manifold and Euclidean space. Specifically, we adopted hyperbolic centroid to aggregate the messages of neighbors:

$$\mathbf{u}_i^{(l+1)} = \textbf{Centroid}(\{\mathbf{u}_i^{(l)}, \{\mathbf{i}_k^{(l)} : k \in N_i\}\}),$$

$$\mathbf{i}_j^{(l+1)} = \textbf{Centroid}(\{\mathbf{i}_j^{(l)}, \{\mathbf{u}_k^{(l)} : k \in N_j\}\}),$$

where $N_i$ denotes the neighbors of node $i$. For a detailed definition of the hyperbolic centroid, see Definition B.7 in Appendix B.

Then the outputs of LHGCN are

$$\mathbf{u}^{local} = \mathbf{u}^{(L)} \text{ and } \mathbf{i}^{local} = \mathbf{i}^{(L)},$$

where $L$ denotes the number of layers of LHGCN.

### 2.3. Hyperbolic Transformer Model

In this section, to address LHGCN's limitations in capturing the global information of the interaction graph and taking into account the unique structure of the bipartite graph in CF, we design a novel hyperbolic cross-attention mechanism for modeling global user-item interactions. Furthermore, since this cross-attention requires operating on all user-item pairs, with computational complexity $O(M \cdot N)$, we propose an approximation approach to reduce the computational complexity to $O(M + N)$.

#### 2.3.1. HYPERBOLIC CROSS-ATTENTION

In CF tasks, there are only interactions between the user set and the item set, which form a bipartite graph. Intuitively, modeling the inner interactions among user groups or item groups would introduce noisy signals and thus deteriorate the performance (Min et al., 2022). For this reason, we introduce a cross-attention mechanism to model only all possible user-item interactions, as detailed in the structure presented in Fig. 2(c). Taking the $i$-th user vector as an example, firstly, the correlation between the $i$-th user vector

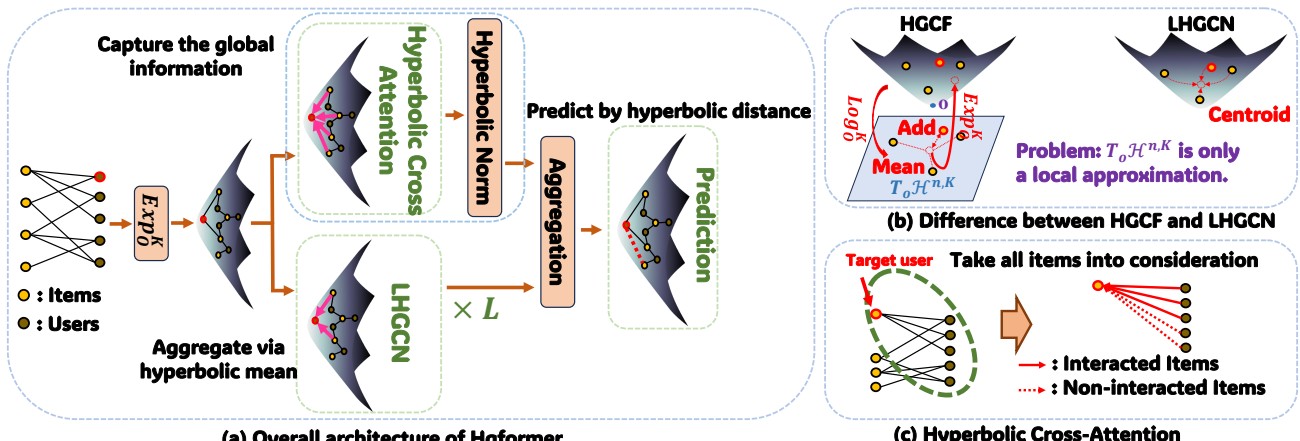

*Figure 2.* Figure(a) shows the overall architecture of Hgformer. We first map the user and item embeddings into a hyperbolic manifold. Then, we use several layers of LHGCN to capture the local information of the interaction graph and use a hyperbolic transformer to capture the overall information of the interaction graph. Finally, we combine these two information and make predictions on the hyperbolic manifold. Figure (b) shows the difference between HGCF and LHGCN in graph convolution. In HGCF, embeddings are mapped from the hyperbolic manifold to the North Pole point's tangent space for aggregation, then back to the hyperbolic manifold, causing information distortion due to the local approximation. LHGCN, on the other hand, aggregates information directly in the hyperbolic manifold. Figure (c) shows the process of hyperbolic cross-attention. Unlike GNN-based models that only consider items a user has interacted with, Hyperbolic Cross-Attention takes both the interacted and non-interacted items into account.

$(i \in \{1, \ldots, N\})$ and the $j$-th item vector $(j \in \{1, \ldots, M\})$ under a specific attention head $h$ is defined as:

$$w_{i,j}^{(h)} = \frac{\exp\left(\mathrm{Sim}(\mathbf{q}_i^{(h)}, \mathbf{k}_j^{(h)})/\tau\right)}{\sum_{l=1}^{M} \exp\left(\mathrm{Sim}(\mathbf{q}_i^{(h)}, \mathbf{k}_l^{(h)})/\tau\right)}, \qquad (1)$$

where $\mathbf{q}_i^{(h)} = \mathbf{W}_Q^{(h)} \otimes^K \mathbf{u}_i$, $\mathbf{k}_i^{(h)} = \mathbf{W}_K^{(h)} \otimes^K \mathbf{i}_i$ and $\otimes^K$ denotes hyperbolic matrix multiplication, which is defined in Definition B.8 in Appendix B. $\tau$ is the temperature parameter. $\mathrm{Sim}(\cdot, \cdot)$ is a function to calculate the similarity of two vectors in the hyperbolic manifold, and it was generally defined as:

$$\mathrm{Sim}(\mathbf{x}, \mathbf{y}) = f(-c_1 d_{\mathcal{M}}^K(\mathbf{x}, \mathbf{y}) + c_2), \qquad (2)$$

where $f(\cdot)$ is a monotonically increasing function, such as exponential maps, linear functions, tanh, sigmoid, etc. and $d_{\mathcal{M}}^K(\cdot, \cdot)$ is the distance function in hyperbolic manifold, which is detailed in Definition B.5 in Appendix B. Then, the representation of the $i$-th user is updated by aggregating all item embeddings with weights $\alpha_{i,j}$:

$$\hat{\mathbf{u}}_i^{(h)} = \sum_{j=1}^{M} w_{i,j}^{(h)} \mathbf{v}_j^{(h)}, \qquad (3)$$

where $\mathbf{v}_j^{(h)} = \mathbf{W}_V^{(h)} \otimes^K \mathbf{i}_j$.

To ensure that the embedding stays in the hyperbolic manifold, we need an extra coefficient $c$ to scale the embedding,

$$\mathbf{u}_i'^{(h)} = c\hat{\mathbf{u}}_i^{(h)}, \qquad (4)$$

where $c = \frac{K}{|\;||\hat{\mathbf{u}}_i^{(h)}||_{\mathcal{M}}|}$ and $K$ is the curvature of the hyperbolic manifold. After that, we aggregate the embeddings of different heads by the hyperbolic centroid defined in Definition. B.7 in Appendix B:

$$\mathbf{u}_i^{global} = \mathbf{Centroid}(\mathbf{u}_i'^{(1)}; \ldots; \mathbf{u}_i'^{(h)}). \qquad (5)$$

We calculate all item embeddings $\mathbf{i}_j^{global}, j \in \{1, \ldots, M\}$ in the same way.

For all the embeddings $\mathbf{x} = [\mathbf{u}^{global}, \mathbf{i}^{global}]$. Finally, for numerical stability, we adopted the definition of Hyperbolic Normalization from (Bdeir et al., 2023) and applied it to normalize the final embeddings $\mathbf{x}$:

$$\mathrm{HN}(x) = \exp_{\beta}^K \left( \mathrm{PT}_{\mathbf{o} \to \beta}^K \left( \gamma \cdot \frac{\mathrm{PT}_{\mu \to 0}^K \left( \log_{\mu}^K(\mathbf{x}) \right)}{\sqrt{\sigma^2 + \epsilon}} \right) \right), \qquad (6)$$

where $\mu = \mathbf{Centroid}(\mathbf{x})$, which is Centroid of $\mathbf{x}$ in hyperbolic manifold and $\sigma^2 = \frac{1}{M+N} \sum_{i=1}^{M+N} (d_{\mathcal{M}}^K(\mathbf{x}_i, \mu))^2$, which is the variance in hyperbolic space, $\beta$ and $\epsilon$ are trainable vectors and $\mathrm{PT}_{\mathbf{o} \to \beta}^K(\cdot)$ means the parallel transport from the point $\mathbf{o}$ to point $\beta$, which is detailed in Definition.B.9 in Appendix B.

### 2.3.2. TOWARDS LINEAR COMPLEXITY

In this section, we introduce the hyperbolic self-attention with only linear computational complexity. Although the above Hyperbolic cross-attention mechanism can model the global interactions between all users and items, its quadratic time complexity prevents its application in real-world sce-

narios when there are numerous users or items. To enable the application of Hyperbolic Transformers to larger datasets, as Fig. 3 shown, the previous complexity of the similarity matrix of hyperbolic vectors is reduced to only $O((M + N)md)$ by our mechanism, where the dimensions $m$ and $d$ are much smaller than $M$ and $N$.

Since the most computationally intensive part of the model is Eq. 3, in this section, our goal is to reduce the computational complexity of Eq. 3 to linear. Firstly, we redefined Eq. 2. Since $K > 0$, $\text{arcosh}(\cdot)$ is a monotonically increasing function, both $\left(-d_{\mathcal{M}}^K(\cdot, \cdot)\right)$ and the Minkowski inner product $\langle \cdot, \cdot \rangle_{\mathcal{M}}$ could be used to compare the similarity between different vectors in hyperbolic space. Then for $\mathbf{x}, \mathbf{y} \in \mathcal{H}^{d+1}$, we redefine the similarity function $\text{Sim}(\cdot, \cdot)$ as the Hyperbolic SoftMax similarity function, $\text{HSM}(\cdot, \cdot) : \mathcal{H}^{d+1,K} \times \mathcal{H}^{d+1,K} \to \mathbb{R}$:

$$\text{HSM}(\mathbf{x}, \mathbf{y}) \triangleq \exp\left(\langle \mathbf{x}, \mathbf{y} \rangle_{\mathcal{M}}\right)$$

then Eq. 3 is redefined as:

$$\hat{\mathbf{u}}_i^{(h)} = \sum_{n=1}^{M} \frac{\exp\left(\left\langle \mathbf{q}_i^{(h)}, \mathbf{k}_j^{(h)} \right\rangle_{\mathcal{M}} / \tau\right)}{\sum_{l=1}^{M} \exp\left(\left\langle \mathbf{q}_i^{(h)}, \mathbf{k}_l^{(h)} \right\rangle_{\mathcal{M}} / \tau\right)} \cdot \mathbf{v}_n^{(h)}, \quad (7)$$

where $\mathbf{q}_i^{(h)}$, $\mathbf{k}_i^{(h)}$ and $\mathbf{v}_i^{(h)}$ follows the settings of Eq. 1. Then, we use an estimation $\widetilde{\kappa}(\mathbf{q}_i^{(h)}, \mathbf{k}_j^{(h)})$ to approximate $\exp\left(\left\langle \mathbf{q}_i^{(h)}, \mathbf{k}_j^{(h)} \right\rangle_{\mathcal{M}}\right)$ in Eq. 7 and we introduce Theorem 4.1, which proves that the aforementioned estimation is an unbiased estimation.

**Theorem 3.1.** *For* $\mathbf{x}, \mathbf{y} \in \mathcal{H}^{d+1,K}$, *with* $\mathbf{x} = (x_0, x_1, \cdots, x_d)^\top$, $\mathbf{y} = (y_0, y_1, \cdots, y_d)^\top$, *and* $\widetilde{\mathbf{x}} = (x_1, x_2, \cdots, x_d)^\top$, $\widetilde{\mathbf{y}} = (y_1, y_2, \cdots, y_d)^\top$, *we have an estimation function* $\widetilde{\kappa}(\cdot, \cdot) : \mathcal{H}^{d+1,K} \times \mathcal{H}^{d+1,K} \to \mathbb{R}$:

$$\widetilde{\kappa}(\mathbf{x}, \mathbf{y}) = \exp\left(\frac{-(x_0 + y_0)^2 + 2K}{2}\right)$$
$$\cdot \mathbb{E}_{\boldsymbol{\omega} \sim \mathcal{N}(\mathbf{0}_d, \mathbf{I}_d)} \left[\exp(\boldsymbol{\omega}^\top(\widetilde{\mathbf{x}} + \widetilde{\mathbf{y}}))\right], \quad (8)$$

*where* $\boldsymbol{\omega} \sim \mathcal{N}(\mathbf{0}_d, I_d)$, $\widetilde{\kappa}(\cdot, \cdot)$ *is an unbiased estimation of the HSM function:*

$$\widetilde{\kappa}(\mathbf{x}, \mathbf{y}) = \text{HSM}(\mathbf{x}, \mathbf{y}). \quad (9)$$

The proof of this theorem is given in Appendix C.1.

Then, such an unbiased estimation function (Eq. 8) can be converted into a dot product of vector functions approximately; the method of converting it is akin to kernel tricks shown as the following lemma:

**Lemma 3.1.** *Define the hyperbolic positive random features* $\phi(\cdot) : \mathcal{H}^{d+1,K} \to \mathbb{R}^m$ :

$$\phi(\mathbf{x}) = \frac{\exp\left(\frac{K - x_0^2}{2}\right)}{\sqrt{m}} \left[\exp\left(\boldsymbol{\omega}_1^\top \widetilde{\mathbf{x}}\right), \cdots, \exp\left(\boldsymbol{\omega}_m^\top \widetilde{\mathbf{x}}\right)\right]$$

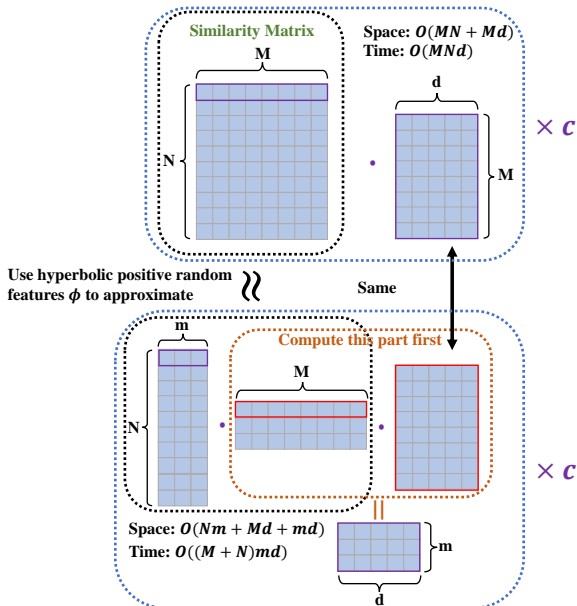

*Figure 3.* This figure shows how we reduce the complexity of hyperbolic graph transformers to linear. We first use $\phi$ to transform the similarity matrix into the multiplication of two smaller matrices and then, we change the order of computation order to reduce the computation complexity.

*where* $\boldsymbol{\omega}_k \sim \mathcal{N}(\mathbf{0}_d, I_d)$ *is i.i.d.,* $m$ *is a constant that could be chosen smaller than* $d$. Then, we have:

$$\phi(\mathbf{x})^\top \phi(\mathbf{y}) \approx \widetilde{\kappa}(\mathbf{x}, \mathbf{y}) = \text{HSM}(\mathbf{x}, \mathbf{y}). \quad (10)$$

The proof of this Lemma is given in Appendix C.2. We adopted a positive random feature map $\phi(\cdot) : \mathcal{H}^{d+1,K} \to \mathbb{R}^m$ to approximate HSM function:

$$\text{HSM}(\mathbf{x}, \mathbf{y})/\tau \approx \phi(\frac{\mathbf{x}}{\sqrt{\tau}})^\top \phi(\frac{\mathbf{y}}{\sqrt{\tau}})) \quad (11)$$

Eq. 11 is proved in Lemma 4.2 and for $\mathbf{x} \in \mathcal{H}^{d+1,K}$, $\mathbf{x} = (x_0, x_1 \cdots x_d)^\top$, $\widetilde{\mathbf{x}} = (x_1, x_2 \cdots x_d)^\top$, the explicit form of positive random feature map with temperature parameter $\tau$ is defined as:

$$\phi(\frac{\mathbf{x}}{\sqrt{\tau}}) = \frac{\exp\left(\frac{K - x_0^2}{2\tau}\right)}{\sqrt{m}} \left[\exp\left(\frac{\boldsymbol{\omega}_1^\top \widetilde{\mathbf{x}}}{\sqrt{\tau}}\right), \cdots, \exp\left(\frac{\boldsymbol{\omega}_m^\top \widetilde{\mathbf{x}}}{\sqrt{\tau}}\right)\right].$$

Then, we can change the computation order and extract common factors by using Eq. 10 to convert the HSM function into the dot product of two feature functions. Then the approximating aggregation function Eq. 12 only has linear complexity and the process is visualized in Fig. 3. Subsequently, the final form of the aggregation function is

proposed as follows:

$$\hat{\mathbf{u}}_i^{(h)} \approx \sum_{j=1}^{M} \frac{\phi\left(\mathbf{q}_i/\sqrt{\tau}\right)^\top \phi\left(\mathbf{k}_j/\sqrt{\tau}\right)}{\sum_{n=1}^{M} \phi\left(\mathbf{q_i}/\sqrt{\tau}\right)^\top \phi\left(\mathbf{k}_n/\sqrt{\tau}\right)} \cdot \mathbf{v}_j$$
$$= \frac{\phi\left(\mathbf{q}_i/\sqrt{\tau}\right)^\top \sum_{j=1}^{M} \phi\left(\mathbf{k}_j/\sqrt{\tau}\right) \cdot (\mathbf{v}_j)^\top}{\phi\left(\mathbf{q}_i/\sqrt{\tau}\right)^\top \sum_{k=1}^{M} \phi\left(\mathbf{k}_n/\sqrt{\tau}\right)}. \quad (12)$$

The error of approximation is bounded and we have:

**Theorem 3.2**: *The error function of approximation*

$$\Delta = \left| \mathrm{HSM}(\mathbf{x}, \mathbf{y}) - \phi(\mathbf{x})^\top \phi(\mathbf{y}) \right|$$

*is bounded by* $\mathcal{O}\left(\sqrt{\frac{\exp(3(\delta - K))}{m\epsilon}}\right)$ *with the probability that:*

$$1 - \epsilon \leqslant \mathbb{P}(\Delta \leqslant \sqrt{\frac{\exp(3(\delta - K))}{m\epsilon}}) \quad (13)$$

*assuming that* $\|\mathbf{x}\|_{\mathcal{E}}^2 \leq \delta$ *,* $\|\mathbf{y}\|_{\mathcal{E}}^2 \leq \delta$ *for* $\mathbf{x}, \mathbf{y} \in \mathcal{H}^{d+1,K}$. The proof is given in Appendix C.3.

Since the upper bound of the error function depends only on the Euclidean norm $\delta$, the curvature constant $K$, the number of positive random features $m$, and the demanding error accuracy $\epsilon$, we can reduce the error by normalizing the vectors and process them in a suitable hyperbolic space, or increase $m$.

### 2.4. Embedding Aggregation and Optimization

**Embedding Aggregation.** To aggregate both structural and global information, we map the embeddings back to Euclidean space using the Log function, and then perform a weighted average:

$$\mathbf{u}^{final} = \mathrm{Exp}_o^K(\alpha \mathrm{Log}_o^K(\mathbf{u}^{global}) + (1 - \alpha) \mathrm{Log}_o^K(\mathbf{u}^{local}))$$
$$\mathbf{i}^{final} = \mathrm{Exp}_o^K(\alpha \mathrm{Log}_o^K(\mathbf{i}^{global}) + (1 - \alpha) \mathrm{Log}_o^K(\mathbf{i}^{local})) \quad (14)$$

where $\alpha$ is a hyperparameter between 0 and 1.

**Prediction.** Margin ranking loss has been extensively used in recommendation tasks (Sun et al., 2021), which separates positive and negative pairs of user items by a given margin. When the gap between a negative and a positive user-item pair exceeds this margin, neither pair contributes to the overall loss, enabling the optimization process to focus on the difficult pairs in the data set. In this work, we use the hyperbolic version of margin-ranking loss as prediction loss. The prediction loss is defined as:

$$L(\mathbf{u}^{final}, \mathbf{i}_{neg}^{final}, \mathbf{i}_{pos}^{final}) = \max\left(d_{\mathcal{M}}(\mathbf{u}^{final}, \mathbf{i}_{pos}^{final})^2\right.$$
$$\left. -d_{\mathcal{M}}(\mathbf{u}^{final}, \mathbf{i}_{neg}^{final})^2 + \lambda, 0\right),$$

where $\lambda$ is a non-negative hyperparameter. $\mathbf{i}_{pos}^{final}$ are the embeddings of the positive samples of this user and $\mathbf{i}_{neg}^{final}$ are the embeddings of negative samples of this user in the same hyperbolic manifold. Positive samples refer to the items that the user has interacted with, while negative samples refer to randomly sampled items that the user has not interacted with.

## 3. Experiments

In this section, we conduct extensive experiments on multiple public datasets to evaluate the proposed method and primarily address the following questions.

**RQ1:** How does Hgformer performs compared to baselines?

**RQ2:** How does each module contributes to the performance?

**RQ3:** Why does Hgformer perform better than other models?

To ensure accuracy and fairness in our experiments, all the models were tested on RecBole (Zhao et al., 2021), a widely accepted framework in the field of recommender systems. The dataset was split into training, validation, and test sets with an 8:1:1 ratio. During training, we adopted an early-stop mechanism for all models, stopping training if no improvement was observed on the validation set for 30 epochs and the best-performing parameters on the validation set were selected for final evaluation on the test set. Due to space limitations, the details of experimental settings are provided in Appendix D, and tail-item analysis and sensitivity analysis are detailed in Appendix E.

### 3.1. Overall Performance Comparison (RQ1)

We compared Hgformer with traditional models, GNN-based models, hyperbolic GNN-based models, and graph transformer-based models across five datasets of different sizes in Table 1. We find the following observations:

1. Across all datasets, Hgformer demonstrates consistent performance improvements. The improvements are particularly significant in the Amazon Book and Amazon CD datasets. On the Amazon Book dataset, compared to the second-best model, HICF, Hgformer achieves relative improvements of 17.6%, 18.9%, 18.1%, and 19.3% in Recall@10, Recall@20, NDCG@10, and NDCG@20, respectively. Similarly, on the Amazon CD dataset, the relative improvements are 18.3%, 15.7%, 15.9%, and 16.9%, respectively.

2. It can be seen that all three hyperbolic GCN-based models show noticeable improvements over traditional GNN models, demonstrating the advantages of hyperbolic GCN in handling recommendation tasks. This is attributed to the fact that the capacity of hyperbolic space

*Table 1.* Overview of performance. N@10 and R@10 are abbreviations for the metrics NDCG@10 and Recall@10, respectively. * represents the significance level $p$-value $< 0.05$. The highest scores for each dataset and metric are emphasized in bold, while the second-best ones are underlined.

| Dataset | Metric | BPR | NGCF | LightGCN | HMLET | HGCF | HICF | HRCF | SGFormer | NodeFormer | Hypformer | Hgformer |
|---|---|---|---|---|---|---|---|---|---|---|---|---|
| **Amazon Book** | **Recall@10** | 0.0357 | 0.0427 | 0.0581 | 0.0347 | 0.0748 | 0.0766 | 0.0745 | 0.0545 | 0.0379 | 0.0376 | **0.0901\*** |
| | **Recall@20** | 0.0538 | 0.0676 | 0.0910 | 0.0553 | 0.1089 | 0.1086 | 0.1080 | 0.0841 | 0.0602 | 0.0605 | **0.1291\*** |
| | **NDCG@10** | 0.0218 | 0.0250 | 0.0344 | 0.0201 | 0.0467 | 0.0485 | 0.0466 | 0.0332 | 0.0222 | 0.0215 | **0.0573\*** |
| | **NDCG@20** | 0.0267 | 0.0317 | 0.0433 | 0.0256 | 0.0560 | 0.0571 | 0.0560 | 0.0412 | 0.0282 | 0.0277 | **0.0681\*** |
| **Amazon CD** | **Recall@10** | 0.0473 | 0.0755 | 0.0770 | 0.0601 | 0.0797 | 0.0826 | 0.0824 | 0.0435 | 0.0454 | 0.0613 | **0.0977\*** |
| | **Recall@20** | 0.0726 | 0.1151 | 0.1174 | 0.0874 | 0.1184 | 0.1211 | 0.1176 | 0.0695 | 0.0710 | 0.0936 | **0.1401\*** |
| | **NDCG@10** | 0.0268 | 0.0426 | 0.0437 | 0.0351 | 0.0451 | 0.0489 | 0.0477 | 0.0242 | 0.0253 | 0.0340 | **0.0567\*** |
| | **NDCG@20** | 0.0334 | 0.0529 | 0.0542 | 0.0421 | 0.0552 | 0.0580 | 0.0577 | 0.0310 | 0.0319 | 0.0424 | **0.0678\*** |
| **Amazon Movie** | **Recall@10** | 0.0585 | 0.0580 | 0.0702 | 0.0491 | 0.0740 | 0.0740 | 0.0761 | 0.0560 | 0.0338 | 0.0294 | **0.0803\*** |
| | **Recall@20** | 0.0929 | 0.0913 | 0.1106 | 0.0817 | 0.1110 | 0.1135 | 0.1157 | 0.0871 | 0.0575 | 0.0528 | **0.1203\*** |
| | **NDCG@10** | 0.0362 | 0.0351 | 0.0440 | 0.0293 | 0.0464 | 0.0465 | 0.0481 | 0.0343 | 0.0200 | 0.0150 | **0.0503\*** |
| | **NDCG@20** | 0.0455 | 0.0442 | 0.0549 | 0.0382 | 0.0566 | 0.0572 | 0.0589 | 0.0428 | 0.0265 | 0.0210 | **0.0612\*** |
| **Douban Book** | **Recall@10** | 0.1059 | 0.1280 | 0.1313 | 0.0926 | 0.1375 | 0.1388 | 0.1357 | 0.0886 | 0.0777 | 0.0446 | **0.1462\*** |
| | **Recall@20** | 0.1588 | 0.1832 | 0.1906 | 0.1440 | 0.1935 | 0.1938 | 0.1892 | 0.1349 | 0.1225 | 0.0772 | **0.2052\*** |
| | **NDCG@10** | 0.0706 | 0.0864 | 0.0922 | 0.0630 | 0.0960 | 0.0968 | 0.0943 | 0.0610 | 0.0484 | 0.0289 | **0.1030\*** |
| | **NDCG@20** | 0.0850 | 0.1013 | 0.1078 | 0.0768 | 0.1113 | 0.1117 | 0.1091 | 0.0734 | 0.0610 | 0.0380 | **0.1189\*** |
| **Douban Movie** | **Recall@10** | 0.0616 | 0.1373 | 0.1339 | 0.1178 | 0.1348 | 0.1370 | 0.1393 | 0.1259 | 0.1118 | 0.0678 | **0.1405\*** |
| | **Recall@20** | 0.0970 | 0.2042 | 0.1989 | 0.1802 | 0.1992 | 0.2034 | 0.2036 | 0.1877 | 0.1722 | 0.1106 | **0.2068\*** |
| | **NDCG@10** | 0.0676 | 0.1256 | 0.1326 | 0.1227 | 0.1248 | 0.1297 | 0.1342 | 0.1263 | 0.1002 | 0.0780 | **0.1322\*** |
| | **NDCG@20** | 0.0725 | 0.1389 | 0.1435 | 0.1325 | 0.1377 | 0.1424 | 0.1456 | 0.1364 | 0.1133 | 0.0845 | **0.1447\*** |
| **Douban Music** | **Recall@10** | 0.1084 | 0.1229 | 0.1258 | 0.0916 | 0.1218 | 0.1276 | 0.1271 | 0.1071 | 0.0791 | 0.0313 | **0.1386\*** |
| | **Recall@20** | 0.1606 | 0.1803 | 0.1802 | 0.1418 | 0.1791 | 0.1843 | 0.1791 | 0.1599 | 0.1239 | 0.0531 | **0.1955\*** |
| | **NDCG@10** | 0.0783 | 0.0899 | 0.0966 | 0.0727 | 0.0927 | 0.0942 | 0.0940 | 0.0842 | 0.0534 | 0.0253 | **0.1024\*** |
| | **NDCG@20** | 0.0917 | 0.1045 | 0.1095 | 0.0840 | 0.1068 | 0.1085 | 0.1070 | 0.0966 | 0.0656 | 0.0306 | **0.1165\*** |

exponentially increases with radius, aligning well with the power-law distributed user-item interaction graph.

3. We also compared our model with three graph transformer models (SGFormer, NodeFormer, and Hypformer). However, since these three models are designed for node classification tasks and compute attention for all node pairs, they tend to overemphasize the relationships between users and between items. This leads to poor performance in recommendation tasks.

### 3.2. Ablation Analysis (RQ2)

We conduct ablation studies on two main components of Hgformer. The results are shown in Fig. 4.We have the following observations:

1. The removal of LHGCN results in the most significant drop in performance. This indicates that LHGCN plays a dominant role in the CF task, and solely using the Hyperbolic Transformer to capture global information between users and items, while ignoring the inherent topological structure of the existing interaction graph, is not sufficient to effectively capture potential user-item relations. Therefore, in recommendation tasks or link prediction tasks, it is difficult to achieve good results by completely abandoning GNNs and relying solely on transformers for prediction. A better strategy for link prediction and recommendation tasks is to use transformers as a supplementary tool.

2. Removing the Hyperbolic Transformer also leads to a remarkable decline in model performance. The Recall@10 metric decreased by 11.6%, 12.9%, and 10.8% on the Amazon Book, Amazon CD, and Douban Music datasets, respectively. Similarly, the NDCG@10 metric experienced reductions of 13.4%, 10.9%, and 7.9% on the Amazon Book, Amazon CD, and Douban Book datasets, respectively. Furthermore, we observed that incorporating the Hyperbolic Transformer led to certain improvements in HGCF, which demonstrates its effectiveness.

3. Replacing LHGCN with HGCF also results in a decline. Through the direct comparison between LHGCN and HGCF, we found that LHGCN outperforms HGCF on the majority of datasets (Amazon Book, Amazon CD, Amazon Movie, Douban Movie). The only exception is the Douban Book dataset, where LHGCN performs slightly worse than HGCF (Recall@10: 0.1368 vs. 0.1375; Recall@20: 0.1916 vs. 0.1935; NDCG@10: 0.0948 vs. 0.0960; NDCG@20: 0.1096 vs. 0.1113). As mentioned in Section 2.2, HGCF requires mapping embeddings back to the tangent space at the North Pole point during message aggregation, which results in information distortion during this process. In contrast, LHGCN performs graph convolution entirely on the hyperbolic manifold.

### 3.3. Case study (RQ3)

In this section, we present a case study to demonstrate the effectiveness of our model in addressing both the local struc-

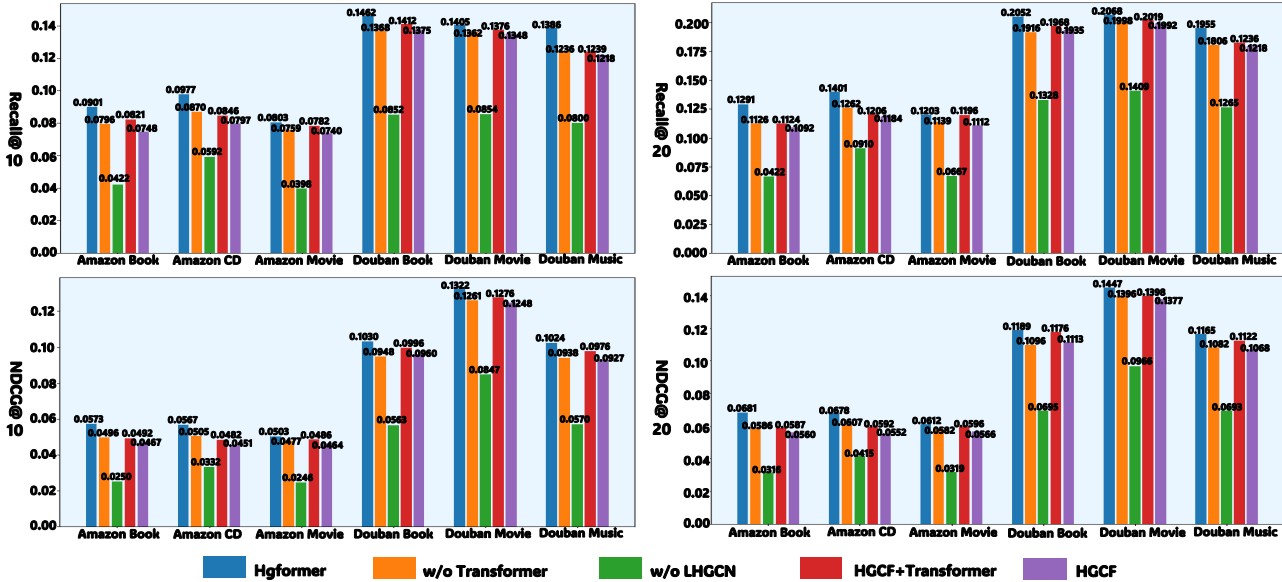

*Figure 4.* To validate the contribution of each module in the model, we individually removed the LHGCN and the Hyperbolic Transformer for evaluation. In the figure, 'w/o LHGCN' and 'w/o Transformer' represent the effects after removing the LHGCN and the Hyperbolic Transformer, respectively. Furthermore, to validate the effectiveness of LHGCN, we also compared LHGCN with HGCF. Specifically, we replaced LHGCN in Hgformer with HGCF and tested the performance and in the figure, it is noted as 'HGCF+Transformer'.

ture modeling problem of GNN-based models and high distortion problem of models in Euclidean space. We selected a user who interacted with 28 items as the subject of the case study. By analyzing items recommended by Hgformer from both head and tail positions, we aim to understand why Hgformer recommended these items while other models did not. It can be observed in Fig 5 that both the Euclidean space models and hyperbolic space models exhibit similar performance for head items, with their rankings being relatively close. However, in the case of tail items, we notice that Euclidean space models tend to rank these items lower, while hyperbolic space models can more effectively identify specific tail items that User_4 prefers. Nonetheless, HGCF fails to recommend items that are far from User_4 in the interaction graph (i.e., items with hops greater than 5). In contrast, Hgformer introduces the Hyperbolic self-attention mechanism, enabling the model to identify distant but relevant items effectively.

## 4. Related Works

**Hyperbolic Neural Networks.** Hyperbolic manifolds, characterized by negative curvature, are effective for modeling hierarchical structures and long-tail distributions (Ravasz & Barabási, 2003). Building on foundational works (Chami et al., 2019; Ganea et al., 2018), various hyperbolic operations—linear layers, activation functions, and graph convolutions—have been introduced. However, many methods map embeddings back to Euclidean space for operations like self-attention and linear transformations. Fully Hyperbolic

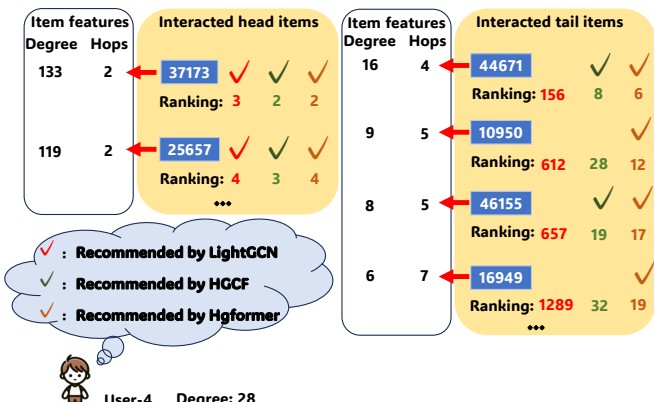

*Figure 5.* We selected user number 4 from the Amazon CD dataset as the subject of analysis. We analyzed the items recommended by Hgformer in the @20 task.

Neural Network (FHNN) (Chen et al., 2021) resolves this by using Lorentzian linear transformations, while H2HGCN (Dai et al., 2021) and LGCN (Zhang et al., 2021b) employ Einstein midpoints and Lorentzian aggregation for fully hyperbolic computations. Despite their efficacy, these methods introduce additional parameters, which are less suitable for collaborative filtering (CF). To address this, we propose LHGCN, a parameter-free, fully hyperbolic graph convolution method tailored for CF.

**Graph Neural Networks for CF.** Graph Neural Networks (GNNs) effectively capture complex user–item relationships, making them widely adopted in CF (Wang et al., 2019a; He et al., 2020; Sun et al., 2021; Chang et al., 2025; Li et al.,

2025; Wang et al., 2025; Xie et al., 2024; Chang et al., 2024; 2023). NGCF (Wang et al., 2019a) enhances expressiveness through multilayer graph convolutions, while LightGCN (He et al., 2020) simplifies GCN architectures by removing non-linear activations and self-loops, improving efficiency and performance. However, traditional GNNs struggle with hierarchical and scale-free graph structures. Hyperbolic GNNs address this limitation: HGCF (Sun et al., 2021) mitigates long-tail issues, HICF (Yang et al., 2022a) balances head and tail recommendations via hyperbolic margin ranking, HRCF (Yang et al., 2022b) incorporates geometric regularization to prevent over-smoothing, and HCTS (Yang et al., 2024b) leverages hyperbolic contrastive learning for cross-domain CF. Despite these advances, existing hyperbolic CF models primarily capture local structures, limiting their ability to model the global user–item graph structure. This highlights the need for more expressive hyperbolic approaches in CF.

**Graph Transformer.** Graph transformers have demonstrated strong representation capabilities for capturing global graph structures and complex interactions (Yun et al., 2019; Ying et al., 2021; Wu et al., 2022; 2023; 2024; Li et al., 2023). However, their high computational complexity restricts their application to small graphs. Recent methods have focused on reducing this complexity, enabling the use of graph transformers in large-scale graphs (Wu et al., 2022; 2024) and recommendation tasks (Li et al., 2023; Chen et al., 2024; Wei et al., 2023; Min et al., 2022). Currently, Hypformer (Yang et al., 2024a) introduces a linear complexity hyperbolic transformer, designed for node classification. The differences between their models with ours can be summerized as follows: We propose LHGCN for effective local structure modeling in hyperbolic space, which is not explored in their work. To achieve linear computational complexity, Hypformer first swaps the multiplication order of space-like values in self-attention vectors and then recalibrates the time-like values. In contrast, we adopt the cross-attention for better performance in CF tasks and directly extends the kernel function to hyperbolic space, which has a theoretical guarantee in approximation errors. Overall, we are the first to propose a hyperbolic graph transformer in the context of collaborative filtering.

## 5. Conclusion

Considering the local structure modeling and embedding distortion issues in previous work of collaborative filtering, we proposed a Hyperbolic Graph Transformer framework, which leverages LHGCN for graph structure modeling and hyperbolic cross-attention for global information modeling. Both theoretical analysis and empirical results demonstrate the superiority of our method, especially in mitigating the long-tail problems in collaborative filtering.

## Acknowledgement

This work was supported by JST SPRING, Grant Number JPMJSP2124.

## Impact Statement

This research addresses the issue of insufficient attention to tail items in recommendation systems. This problem is related to fairness from the perspective of merchants. The model proposed in this study ensures that while maintaining recommendation accuracy for users, merchants' items also receive sufficient attention, thereby ensuring fairness.

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

# Appendix

## A. Notations

We summerize all the notations of this paper as follows:

Table 2. Notations used in this paper

| Notation | Description |
|---|---|
| $\mathcal{V}_u$ & $\mathcal{V}_i$ | The user and item set |
| $N$ | Number of users |
| $M$ | Number of items |
| $N_i$ | The neighbors of node $i$ |
| $\mathcal{E}$ | Interactions between users and items |
| $\Theta$ | Model parameters |
| $d$ | Dimension of latent embeddings. |
| $\mathbf{u}^{\mathcal{E}}$ & $\mathbf{i}^{\mathcal{E}}$ | User&item embeddings in Euclidean space. |
| $\mathbf{u}$ & $\mathbf{i}$ | User&item embeddings in hyperbolic manifold. |
| $\mathbf{u}_k$ & $\mathbf{i}_l$ | $k$-th and $l$-th vector of $\mathbf{u}$ & $\mathbf{i}$ |
| $\mathbf{u}^{(l)}$ & $\mathbf{i}^{(l)}$ | User&item embeddings after $l$-th layer of LHGCN |
| $\mathbf{q}_i^{(h)}$ & $\mathbf{k}_j^{(h)}$ & $\mathbf{v}_j^{(h)}$ | $i$-th query, $j$-th key and value of $h$-head |
| $\mathbf{u}^{local}$ & $\mathbf{i}^{local}$ | User&item embeddings after LHGCN |
| $\mathbf{u}^{global}$ & $\mathbf{i}^{global}$ | User&item embeddings of hyperbolic cross attention. |
| $w_{i,j}^{(h)}$ | Similarity score of $i$-th user and $j$-item of $h$-th head in the self-attention |
| $\hat{\mathbf{u}}_i^{(h)}$ | $i$-th user embedding of $h$-th head after Euclidean weighted sum. |
| $\mathbf{u}_i'^{(h)}$ | $i$-th user embedding of $h$-th head after hyperbolic weighted sum. |
| $m$ | Number of random features. |

## B. Preliminaries of Hyperbolic Geometry

In this section, we summarize key concepts from hyperbolic geometry that form the foundation of our method.

**Definition B.1** (Minkowski Inner Product). Let

$$\langle \cdot, \cdot \rangle_{\mathcal{M}} \colon \mathbb{R}^{n+1} \times \mathbb{R}^{n+1} \to \mathbb{R}$$

be the bilinear map defined by

$$\langle \mathbf{u}, \mathbf{v} \rangle_{\mathcal{M}} := -u_0 v_0 + \sum_{i=1}^{n} u_i v_i \;=\; \mathbf{u}^{\top} \mathbf{J} \, \mathbf{v},$$

where $\mathbf{J} = \mathrm{diag}(-1, 1, \ldots, 1) \in \mathbb{R}^{(n+1) \times (n+1)}$. Although $\langle \cdot, \cdot \rangle_{\mathcal{M}}$ is not a true inner product (because $\mathbf{J}$ has a negative eigenvalue), it is commonly referred to as the *Minkowski (pseudo) inner product*.

For any $K > 0$, the condition

$$\langle \mathbf{x}, \mathbf{x} \rangle_{\mathcal{M}} = -K$$

i.e.,

$$x_0^2 \; = \; K + \sum_{i=1}^{n} x_i^2 \; \geq \; K,$$

this defines two connected components, determined by the sign of $x_0$. Note that $x_0 \geq K$ or $x_0 \leq -K$. The condition $x_0 > 0$ selects one of them.

**Definition B.2** (Hyperbolic Manifold). Consider the following subset of $\mathbb{R}^{n+1}$:

$$\mathcal{H}^{n,K} \; := \; \{ \mathbf{x} \in \mathbb{R}^{n+1} : \langle \mathbf{x}, \mathbf{x} \rangle_{\mathcal{M}} = -K \text{ and } x_0 > 0 \}.$$

This is often called $n$-dimensional Hyperbolic Manifold with curvature $-\frac{1}{K}$.

The defining function $h(\mathbf{x}) \; = \; \langle \mathbf{x}, \mathbf{x} \rangle_{\mathcal{M}} + K$ has differential $\mathrm{D}h(\mathbf{x})[\mathbf{u}] \; = \; 2\langle \mathbf{x}, \mathbf{u} \rangle_{\mathcal{M}} \; = \; (2\,\mathbf{J}\,\mathbf{x})^{\mathsf{T}}\mathbf{u}$.

Since $\mathbf{J}$ is invertible and $\mathbf{J}\,\mathbf{x} = 0$ if and only if $\mathbf{x} = 0$, we have $\mathrm{D}h(\mathbf{x}) \neq \mathbf{0}$ for all $\mathbf{x} \in \mathcal{H}^{n,K}$. Hence, $\mathrm{D}h(\mathbf{x})$ is surjective, implying $\mathcal{H}^{n,K}$ is a smooth embedded submanifold of $\mathbb{R}^{n+1}$ of dimension $n$ (see, e.g., (Boumal, 2023, Definition 3.10 & Theorem 3.15)).

**Definition B.3** (Tangent Space). For any $\mathbf{x} \in \mathcal{H}^{n,K}$, the tangent space is given by

$$T_{\mathbf{x}}\mathcal{H}^{n,K} \; = \; \ker \mathrm{D}h(\mathbf{x}) \; = \; \{ \mathbf{u} \in \mathbb{R}^{n+1} : \langle \mathbf{x}, \mathbf{u} \rangle_{\mathcal{M}} = 0 \} \; = \; \{ \mathbf{u} \in \mathbb{R}^{n+1} : x_0\,u_0 = \sum_{i=1}^{n} x_i\,u_i \}.$$

Restricting the Minkowski product $\langle \cdot, \cdot \rangle_{\mathcal{M}}$ to each $T_{\mathbf{x}}\mathcal{H}^{n,K}$ provides a Riemannian metric on $\mathcal{H}^{n,K}$. Thus, $\mathcal{H}^{n,K}$ becomes a Riemannian manifold with constant negative sectional curvature equal to $-\frac{1}{K}$.

**Definition B.4** (North Pole). Define the *north pole* of $\mathcal{H}^{n,K}$ by

$$\mathbf{o} := \left( \sqrt{K}, 0, \ldots, 0 \right) \in \mathcal{H}^{n,K}.$$

Its tangent space satisfies

$$T_{\mathbf{o}}\mathcal{H}^{n,K} \; = \; \{ \mathbf{u} \in \mathbb{R}^{n+1} : \langle \mathbf{o}, \mathbf{u} \rangle_{\mathcal{M}} = 0, \; u_0 = 0 \} \; = \; \{ (0, \mathbf{u}') : \mathbf{u}' \in \mathbb{R}^n \} \; \cong \; \mathbb{R}^n.$$

For a fixed dimension $n$, different curvatures $-\frac{1}{K}$ yield different manifolds $\mathcal{H}^{n,K}$, each with its own north pole $\mathbf{o}$ but all sharing the same tangent space structure at $\mathbf{o}$.

**Definition B.5** (Hyperbolic Distance). The distance function on $\mathcal{H}^{n,K}$ induced by the Minkowski product is

$$d_{\mathcal{M}}^{K}(\mathbf{x}, \mathbf{y}) \; = \; \sqrt{K} \, \mathrm{arcosh}\!\left( -\frac{\langle \mathbf{x}, \mathbf{y} \rangle_{\mathcal{M}}}{K} \right),$$

for all $\mathbf{x}, \mathbf{y} \in \mathcal{H}^{n,K}$.

**Well-definedness of Hyperbolic Distance:**

We define two vectors in $\mathbb{R}^n$: $x' \; := \; \left( \sqrt{K}, x_1, \ldots, x_n \right), \quad y' \; := \; \left( \sqrt{K}, y_1, \ldots, y_n \right)$.

Since $\mathbf{x}, \mathbf{y} \in \mathcal{H}^{n,K}$ implies $\|\mathbf{x}'\|_2 = x_0 > 0$ and $\|\mathbf{y}'\|_2 = y_0 > 0$, we have

$$-\langle \mathbf{x}, \mathbf{y} \rangle_{\mathcal{M}} \; = \; x_0\,y_0 \; - \; \sum_{i=1}^{n} x_i\,y_i \; = \; \|\mathbf{x}'\|_2\,\|\mathbf{y}'\|_2 \; - \; \sum_{i=1}^{n} x_i\,y_i \; + \; K \; = \; \|\mathbf{x}'\|_2\,\|\mathbf{y}'\|_2 \; - \; \langle \mathbf{x}', \mathbf{y}' \rangle \; + \; K,$$

where $\langle \cdot, \cdot \rangle$ on the right is the usual inner product in $\mathbb{R}^n$.

By the Cauchy–Schwarz inequality,

$$\|\mathbf{x}'\|_2\,\|\mathbf{y}'\|_2 \; - \; \langle \mathbf{x}', \mathbf{y}' \rangle \; \geq \; 0 \quad \Longrightarrow \quad -\langle \mathbf{x}, \mathbf{y} \rangle_{\mathcal{M}} \; \geq \; K \quad \Longrightarrow \quad -\frac{\langle \mathbf{x}, \mathbf{y} \rangle_{\mathcal{M}}}{K} \; \geq \; 1.$$

Hence, $\mathrm{arcosh}\!\left(-\frac{\langle \mathbf{x}, \mathbf{y} \rangle_{\mathcal{M}}}{K}\right)$ is well-defined, making $d_{\mathcal{M}}^K(\mathbf{x}, \mathbf{y})$ well-defined for all $\mathbf{x}, \mathbf{y} \in \mathcal{H}^{n,K}$.

**Definition B.6** (Exponential and Logarithmic Maps). For $\mathbf{x} \in \mathcal{H}^{n,K}$ and $\mathbf{v} \in T_{\mathbf{x}}\mathcal{H}^{n,K}$ with $\mathbf{v} \neq \mathbf{0}$, the exponential map is

$$\mathrm{Exp}_{\mathbf{x}}^K(\mathbf{v}) \; = \; \cosh\!\left(\frac{\|\mathbf{v}\|_{\mathcal{M}}}{\sqrt{K}}\right) \mathbf{x} \; + \; \sqrt{K}\, \sinh\!\left(\frac{\|\mathbf{v}\|_{\mathcal{M}}}{\sqrt{K}}\right) \frac{\mathbf{v}}{\|\mathbf{v}\|_{\mathcal{M}}},$$

and for $\mathbf{y} \in \mathcal{H}^{n,K}$ with $\mathbf{y} \neq \mathbf{x}$, the logarithmic map is

$$\mathrm{Log}_{\mathbf{x}}^K(\mathbf{y}) \; = \; \frac{d_{\mathcal{M}}^K(\mathbf{x}, \mathbf{y})}{\|\mathrm{Proj}_{\mathbf{x}}(\mathbf{y})\|_{\mathcal{M}}} \, \mathrm{Proj}_{\mathbf{x}}(\mathbf{y}).$$

**Example: Mapping $\mathbb{R}^n$ to $\mathcal{H}^{n,K}$.** Following (Chami et al., 2019), let $\mathbf{x}^{\mathcal{E}} \in \mathbb{R}^n$ be a Euclidean vector and $\mathbf{o} := (\sqrt{K}, 0, \ldots, 0)$ be the north pole in $\mathcal{H}^{n,K}$. We interpret $(0, \mathbf{x}^{\mathcal{E}})$ as a vector in $T_{\mathbf{o}}\mathcal{H}^{n,K}$. Then, using the exponential map,

$$\mathbf{x}^{\mathcal{H}} \; := \; \exp_{\mathbf{o}}^K\big((0, \mathbf{x}^{\mathcal{E}})\big) \; = \; \left(\sqrt{K}\, \cosh\big(\tfrac{\|\mathbf{x}^{\mathcal{E}}\|_2}{\sqrt{K}}\big), \; \sqrt{K}\, \sinh\big(\tfrac{\|\mathbf{x}^{\mathcal{E}}\|_2}{\sqrt{K}}\big) \frac{\mathbf{x}^{\mathcal{E}}}{\|\mathbf{x}^{\mathcal{E}}\|_2}\right).$$

**Definition B.7** (Hyperbolic Centroid). Let $\mathcal{N} \subset \mathcal{H}^{d+1,K}$ be a finite set of points $\{\mathbf{x}_i\}$ with corresponding weights $\{w_i\}$. The *weighted centroid* $\mathbf{c} \in \mathcal{H}^{d+1,K}$ is defined as the minimizer of

$$\min_{\mathbf{c} \in \mathcal{H}^{d+1,K}} \; \sum_{\mathbf{x}_i \in \mathcal{N}} w_i \big(d_{\mathcal{M}}^K(\mathbf{c}, \mathbf{x}_i)\big)^2.$$

A closed-form solution (cf. (Chami et al., 2019)) is

$$\mathbf{Centroid}(\mathcal{N}, \mathbf{w}) \; = \; \sqrt{K}\, \frac{\sum_{\mathbf{x}_i \in \mathcal{N}} w_i\, \mathbf{x}_i}{\left\|\sum_{\mathbf{x}_i \in \mathcal{N}} w_i\, \mathbf{x}_i\right\|_{\mathcal{M}}}.$$

When all weights are equal, we have

$$\mathbf{Centroid}(\mathcal{N}) \; = \; \sqrt{K}\, \frac{\sum_{\mathbf{x}_i \in \mathcal{N}} \mathbf{x}_i}{\left\|\sum_{\mathbf{x}_i \in \mathcal{N}} \mathbf{x}_i\right\|_{\mathcal{M}}}.$$

**Definition B.8** (Hyperbolic Matrix Multiplication). Let $\mathbf{W}$ be a matrix in $\mathbb{R}^{n \times n}$ and $\mathbf{x} \in \mathcal{H}^{n,K}$. Following (Chami et al., 2019), the *hyperbolic matrix multiplication* is

$$\mathbf{W} \otimes^K \mathbf{x} \; := \; \mathrm{Exp}_{\mathbf{o}}^K\!\Big(\mathbf{W}\, \mathrm{Log}_{\mathbf{o}}^K(\mathbf{x})\Big),$$

where $\mathbf{o}$ is the north pole.

**Definition B.9** (Parallel Transport). If $\mathbf{x}, \mathbf{y} \in \mathcal{H}^{n,K}$ are connected by a unique geodesic, then the *parallel transport* of $\mathbf{v} \in T_{\mathbf{x}}\mathcal{H}^{n,K}$ to $T_{\mathbf{y}}\mathcal{H}^{n,K}$ along this geodesic is

$$P_{\mathbf{x} \to \mathbf{y}}(\mathbf{v}) \; = \; \mathbf{v} \; - \; \frac{\langle \log_{\mathbf{x}}(\mathbf{y}), \mathbf{v} \rangle_{\mathcal{M}}}{\big(d_{\mathcal{M}}^K(\mathbf{x}, \mathbf{y})\big)^2} \Big(\log_{\mathbf{x}}(\mathbf{y}) + \log_{\mathbf{y}}(\mathbf{x})\Big).$$

These definitions equip us with the fundamental tools for hyperbolic geometry. They serve as the basis for our approach described in this paper.

# C. Proof of Theorems

## C.1. Proof of Theorem 3.1

**Proof.**

We set $\boldsymbol{\omega} \sim \mathcal{N}(\mathbf{0}_d, \mathbf{I}_d) \in \mathbb{R}^d$. For any $\mathbf{x} \in \mathbb{R}^d$ we have

$$(2\pi)^{-d/2} \int_{\boldsymbol{\omega}} \exp\left(-\|\boldsymbol{\omega} - \mathbf{x}\|_{\mathcal{E}}^2/2\right) d\boldsymbol{\omega} = 1 \tag{15}$$

and for any $\mathbf{x} \in \mathcal{H}^{d+1} : -x_0^2 + x_1^2 + \cdots x_d^2 = -K$

Then by definition of HSM function, we have:

$$\mathrm{HSM}(\mathbf{x}, \mathbf{y}) = \exp\left(\langle \mathbf{x}, \mathbf{y} \rangle_{\mathcal{M}}\right) \tag{16}$$

$$= \exp(-x_0 \cdot y_0) \exp\left(-\|\widetilde{\mathbf{x}}\|_{\mathcal{E}}^2/2\right) \exp\left(\|\widetilde{\mathbf{x}} + \widetilde{\mathbf{y}}\|_{\mathcal{E}}^2/2\right) \cdot \exp\left(-\|\widetilde{\mathbf{y}}\|_{\mathcal{E}}^2/2\right) \tag{17}$$

multiple 1 and utilize Eq. 15, we get:

$$= \exp\left(-(2x_0 y_0 + \sum_{i=1}^{d} x_i^2 + \sum_{i=1}^{d} y_i^2)/2\right) (2\pi)^{-d/2} \cdot \exp\left(\|\widetilde{\mathbf{x}} + \widetilde{\mathbf{y}}\|_{\mathcal{E}}^2/2\right) \int_{\boldsymbol{\omega}} \exp\left(-\|\boldsymbol{\omega} - (\widetilde{\mathbf{x}} + \widetilde{\mathbf{y}})\|_{\mathcal{E}}^2/2\right) d\boldsymbol{\omega} \tag{18}$$

expand it inside:

$$= \exp\left(\frac{-(2x_0 y_0 + x_0^2 - K + y_0^2 - K)}{2}\right) (2\pi)^{-d/2} \int_{\boldsymbol{\omega}} \exp(-\|\boldsymbol{\omega}\|_{\mathcal{E}}^2/2 + \boldsymbol{\omega}^\top (\widetilde{\mathbf{x}} + \widetilde{\mathbf{y}}) - \|\widetilde{\mathbf{x}} + \widetilde{\mathbf{y}}\|_{\mathcal{E}}^2/2 + \|\widetilde{\mathbf{x}} + \widetilde{\mathbf{y}}\|_{\mathcal{E}}^2/2) d\boldsymbol{\omega} \tag{19}$$

simplify the expression:

$$= \exp\left(\frac{-(x_0 + y_0)^2 + 2K}{2}\right) (2\pi)^{-d/2} \int_{\boldsymbol{\omega}} \exp\left(-\|\boldsymbol{\omega}\|_{\mathcal{E}}^2/2 + \boldsymbol{\omega}^\top (\widetilde{\mathbf{x}} + \widetilde{\mathbf{y}})\right) d\boldsymbol{\omega} \tag{20}$$

show this integration in a form similar to Eq. 15

$$= \exp\left(\frac{-(x_0 + y_0)^2 + 2K}{2}\right) (2\pi)^{-d/2} \int_{\boldsymbol{\omega}} \exp\left(-\|\boldsymbol{\omega}\|_{\mathcal{E}}^2/2\right) \cdot \exp\left(\boldsymbol{\omega}^\top \widetilde{\mathbf{x}}\right) \cdot \exp\left(\boldsymbol{\omega}^\top \widetilde{\mathbf{y}}\right) d\boldsymbol{\omega} \tag{21}$$

So it is the expectation of $\left[\exp(\boldsymbol{\omega}^\top (\widetilde{\mathbf{x}} + \widetilde{\mathbf{y}})\right]$:

$$= \exp\left(\frac{-(x_0 + y_0)^2 + 2K}{2}\right) \mathbb{E}_{\boldsymbol{\omega} \sim \mathcal{N}(\mathbf{0}, \mathbf{I}_d)} \left[\exp(\boldsymbol{\omega}^\top (\widetilde{\mathbf{x}} + \widetilde{\mathbf{y}})\right] \tag{22}$$

which shows that is an unbiased estimation of HSM function. $\qquad \square$

## C.2. Proof of Lemma 3.1

**Proof.**

By definition of function $\phi(\mathbf{x})$:

$$\phi(\mathbf{x})^\top \phi(\mathbf{y}) = \exp(\frac{2K - x_0^2 - y_0^2}{2}) \frac{1}{m} \left[ \exp\left(\boldsymbol{\omega}_1^\top (\widetilde{\mathbf{x}} + \widetilde{\mathbf{y}})\right), \cdots, \exp\left(\boldsymbol{\omega}_m^\top (\widetilde{\mathbf{x}} + \widetilde{\mathbf{y}})\right) \right] \tag{23}$$

since the error between $\exp(2K - x_0^2 - y_0^2/2)$ and $\exp\left(-(x_0 + y_0)^2 + 2K/2\right)$ are small, and the remaining part is the sampling function of the expectation, then we can take the approximation:

$$\approx \exp\left(\frac{-(x_0 + y_0)^2 + 2K}{2}\right) \mathbb{E}_{\boldsymbol{\omega} \sim \mathcal{N}(\mathbf{0}_d, \mathbf{I}_d)} \left[\exp(\boldsymbol{\omega}^\top (\widetilde{\mathbf{x}} + \widetilde{\mathbf{y}})\right] \tag{24}$$

which equals to $\widetilde{\kappa}(\mathbf{x}, \mathbf{y})$, and by Eq. 8, it also equals to $\mathrm{HSM}(\mathbf{x}, \mathbf{y})$, then complete the proof.

$\square$

## C.3. Proof of Theorem 3.2

**Proof.**

First, by the definition of error function:

$$\Delta(\mathbf{x}, \mathbf{y}) = \left| \exp\left(\langle \mathbf{x}, \mathbf{y} \rangle_{\mathcal{M}}\right) - \phi(\mathbf{x})^\top \phi(\mathbf{y}) \right| \tag{25}$$

take an expansion:

$$\Delta(\mathbf{x}, \mathbf{y}) = \left| \exp\left(\frac{2K - (x_0 + y_0)^2}{2}\right) \exp\left(\widetilde{\mathbf{x}}^\top \widetilde{\mathbf{y}}\right) - \exp\left(\frac{2K - x_0^2 - y_0^2}{2}\right) \frac{1}{m} \sum_{i=1}^m \exp(\boldsymbol{\omega}_i^\top (\widetilde{\mathbf{x}} + \widetilde{\mathbf{y}})) \right| \tag{26}$$

we set

$$a = \exp\left(-x_0 y_0\right) \tag{27}$$

$$X = \exp\left(\frac{2K - x_0^2 - y_0^2}{2}\right) \frac{1}{m} \sum_{i=1}^m \exp(\boldsymbol{\omega}_i^\top (\widetilde{\mathbf{x}} + \widetilde{\mathbf{y}})) \tag{28}$$

then we have:

$$\mathbb{E}(X) = \exp\left(\frac{2K - x_0^2 - y_0^2}{2}\right) \mathbb{E}_{\boldsymbol{\omega} \sim \mathcal{N}(\mathbf{0}, \mathbf{I}_d)} \left[\exp\left(\boldsymbol{\omega}^\top (\widetilde{\mathbf{x}} + \widetilde{\mathbf{y}})\right)\right] \tag{29}$$

By Markov's inequality:

$$\mathbb{P}\left(\Delta \leqslant \sqrt{\frac{\exp(3(\delta - K))}{m\epsilon}}\right) \geqslant 1 - \frac{\left[\mathbb{E}\left((X - a\mathbb{E}(X))^2\right)\right] m\epsilon}{\exp(3(\delta - K))} \tag{30}$$

take an expansion:

$$= 1 - \frac{\left[\mathbb{E}\left(X^2\right) + \left(a^2 - 2a\right)\mathbb{E}^2(X)\right]m\epsilon}{\exp(3(\delta - K))} \tag{31}$$

plug in the expectation:

$$= 1 - \left[\frac{1}{m}\exp(\|\widetilde{\mathbf{x}} + \widetilde{\mathbf{y}}\|_{\mathcal{E}}^2)\exp(2\widetilde{\mathbf{x}}^\top\widetilde{\mathbf{y}})\right.$$
$$\left.(1 + (a^2 - 2a)\exp(-\|\widetilde{x} + \widetilde{y}\|_{\mathcal{E}}^2))\right]m\epsilon / \left(\exp(3(\delta - K))\right) \tag{32}$$

Since $a = \exp\left(-x_0 y_0\right) \leqslant e^{-K}$, $K \in (0, \infty)$, we get:

$$\geqslant 1 - \epsilon\exp\left(\|\widetilde{\mathbf{x}} + \widetilde{\mathbf{y}}\|_{\mathcal{E}}^2 + 2\widetilde{\mathbf{x}}^\top\widetilde{\mathbf{y}} - 3(\delta - K)\right) \tag{33}$$

We assume that $\|\mathbf{x}\|_{\mathcal{E}}^2 \leq \delta, \|\mathbf{y}\|_{\mathcal{E}}^2 \leq \delta$, then

$$\|\widetilde{\mathbf{x}}\|_{\mathcal{E}}^2 \leq \frac{\delta - K}{2}, \|\widetilde{\mathbf{y}}\|_{\mathcal{E}}^2 \leq \frac{\delta - K}{2} \tag{34}$$

and we also have:

$$\|\widetilde{\mathbf{x}} + \widetilde{\mathbf{y}}\|_{\mathcal{E}}^2 \leqslant 2(\delta - K), \widetilde{\mathbf{x}}^\top\widetilde{\mathbf{y}} \leqslant \frac{\delta - K}{2}, \tag{35}$$

then the last inequality holds:

$$1 - \epsilon\exp\left(\|\widetilde{\mathbf{x}} + \widetilde{\mathbf{y}}\|_{\mathcal{E}}^2 + 2\widetilde{\mathbf{x}}^\top\widetilde{\mathbf{y}} - 3(\delta - K)\right) \geqslant 1 - \epsilon \tag{36}$$

Finally, we prove the upper bound of the error with probability :

$$\mathbb{P}\left(\Delta \leqslant \sqrt{\frac{\exp(3(\delta - K))}{m\epsilon}}\right) \geqslant 1 - \epsilon \tag{37}$$

which completes the proof.

$\square$

## D. Details of Experimental Setup

In this section, we will describe the detailed settings of our experiments.

### D.1. Baselines

To demonstrate the effectiveness of our proposed model, we compare our model with four categories of models: (A) Traditional models BPR(Rendle et al., 2012) and DMF(Xue et al., 2017). (B) GNN-based models NGCF(Wang et al., 2019b), LightGCN(He et al., 2020) and HMLET. (C) Hyperbolic GNN-based methods HGCF(Sun et al., 2021), HRCF(Yang et al., 2022b) and HICF(Yang et al., 2022a). (D) Graph Transformer-based methods SGFormer(Wu et al., 2024), NodeFormer(Wu et al., 2022) and Hypformer(Yang et al., 2024b). Specifically, BPR, DMF, NGCF, LightGCN, and HMLET utilize the RecBole implementations, while HGCF, HRCF, HICF, SGFormer, NodeFormer, and Hypformer are migrated from their original implementations into RecBole.

We introduce the baseline models in detail as follows:

- **BPR** (Rendle et al., 2012) is a foundational matrix factorization approach for recommendation. It optimizes a pairwise ranking objective that directly models user preferences by enforcing higher scores for the positively interacted items than for the negative ones.

- **NGCF** (Wang et al., 2019b) is a GCN-based recommendation model that exploits high-order user–item interactions by iteratively propagating embeddings through the user–item graph, thus capturing collaborative signals beyond direct neighbors.

- **LightGCN** (He et al., 2020) is a simplified GCN model for recommendation that removes feature transformations and non-linear activation functions, focusing purely on neighborhood aggregation to reduce complexity while maintaining strong performance.

- **HMLET** (Kong et al., 2022) combines both linear and non-linear propagation layers to capture various aspects of user–item relationships, yielding improved performance on general recommendation tasks.

- **HGCF** (Sun et al., 2021) extends graph-based recommendation into hyperbolic geometry, aiming to better model hierarchical or tree-like structures in the user–item graph.

- **HRCF** (Yang et al., 2022b) is also a hyperbolic graph neural network for recommendation, which incorporates geometric regularization to further enhance representation learning in hyperbolic space.

- **HICF** (Yang et al., 2022a) leverages a margin ranking loss in hyperbolic space, explicitly exploiting geometric properties for more accurate recommendation performance.

- **SGFormer** (Wu et al., 2024) is a linear graph transformer model designed for large-scale graphs, utilizing a kernel-like attention mechanism to reduce the quadratic complexity typically associated with transformers.

- **NodeFormer** (Wu et al., 2022) is another kernel-based graph transformer model that achieves scalability through linearized attention computations, making it suitable for large and dense graphs.

- **Hypformer** (Yang et al., 2024a) is a hyperbolic graph transformer model, bringing transformer-style attention into hyperbolic space to better capture hierarchical structures and geometric relationships.

### D.2. Datasets

To validate the effectiveness of the model, we conducted experiments on six datasets with varying sizes and densities, including Amazon datasets[1] (Amazon Book, Amazon CD, and Amazon Movie), as well as Douban datasets[2] (Douban Book, Douban Movie, and Douban Music). Detailed statistics of the datasets are summarized in Table 3.

*Table 3.* Dataset Statistics

| Metric | Amazon-Book | Amazon-CD | Amazon-Movie | Douban-Book | Douban-Movie | Douban-Music |
|---|---|---|---|---|---|---|
| #User | 211,170 | 66,317 | 26,969 | 18,086 | 22,041 | 15,996 |
| #Item | 163,789 | 58,869 | 18,564 | 33,068 | 25,802 | 39,749 |
| #Interactions | 5,069,747 | 952,547 | 762,957 | 809,248 | 2,553,305 | 1,116,984 |
| Density | 0.015% | 0.024% | 0.150% | 0.140% | 0.459% | 0.176% |

### D.3. Definition of evaluation metrics

We used two metrics for the evaluation. The first metric is Recall@K, which measures the fraction of relevant items retrieved out of all relevant items. The second metric is NDCG@K, which is a measure of ranking quality in which positions are discounted logarithmically. It accounts for the position of the hits by assigning higher scores to hits at the top ranks. The formal definition is given as follows: We employed two metrics for evaluation. i) Recall@K, which measures the fraction of relevant items retrieved out of all relevant items, which is formally defined as:

---

[1] https://jmcauley.ucsd.edu/data/amazon
[2] https://www.douban.com

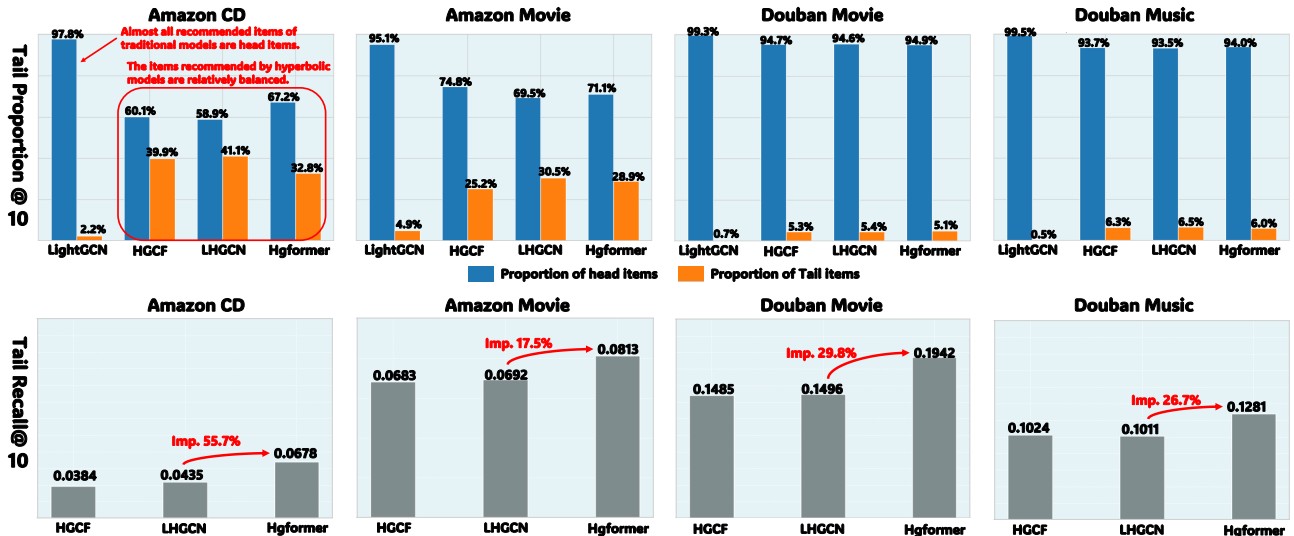

*Figure 6.* We define the top 20% of items in terms of popularity as head items and the rest as tail items. We calculated the proportion of head items and tail items among all recommended items for each model. To further analyze the performance of Hgformer on tail items, we evaluate the performance of three hyperbolic-based models on tail items.

$$\text{Recall@K} = \frac{1}{|\mathcal{U}|} \sum_{u \in \mathcal{U}} \frac{|\hat{R}(u) \cap R(u)|}{|R(u)|}, \tag{38}$$

where $\mathcal{U}$ is the set of all users, $\hat{R}(u)$ represent a ranked list of items that a model produces, and $R(u)$ represent a ground-truth set of items that user has interacted with. ii) NDCG@K, which is a measure of ranking quality where positions are discounted logarithmically. It accounts for the position of the hits by assigning higher scores to hits at top ranks and it is formally defined as:

$$\text{NDCG@K} = \frac{1}{|\mathcal{U}|} \sum_{u \in \mathcal{U}} \left( \frac{1}{\sum_{i=1}^{\min(|R(u)|,K)} \frac{1}{\log_2(i+1)}} \sum_{i=1}^{K} \frac{\delta(i \in R(u))}{\log_2(i+1)} \right), \tag{39}$$

where $\delta(\cdot)$ is an indicator function. In this research, we set K as 10 and 20.

## E. Extra experiments

We implement Sensitivity Analysis and Analysis on Tail Items to address the following questions:

**RQ4:** How well does Hgformer perform on the head and tail items?

**RQ5:** Is Hgformer sensitive to different hyperparameters?

### E.1. Tail-item analysis(RQ4)

In this section, we analyze the results on tail items to demonstrate Hgformer's capability to mitigate long-tail issues.

#### E.1.1. TAIL PERCENTAGE ANALYSIS

We calculate the proportion of tail items recommended by each model and tail items are defined as those whose popularity ranks in the last 80%. For this purpose, we designed a metric called tail percentage, which is formally defined as follows:

$$\text{TailPercentage@K} = \frac{1}{|U|} \sum_{u \in U} \frac{\sum_{i \in R_u} \delta(i \in T)}{|R_u|}, \tag{40}$$

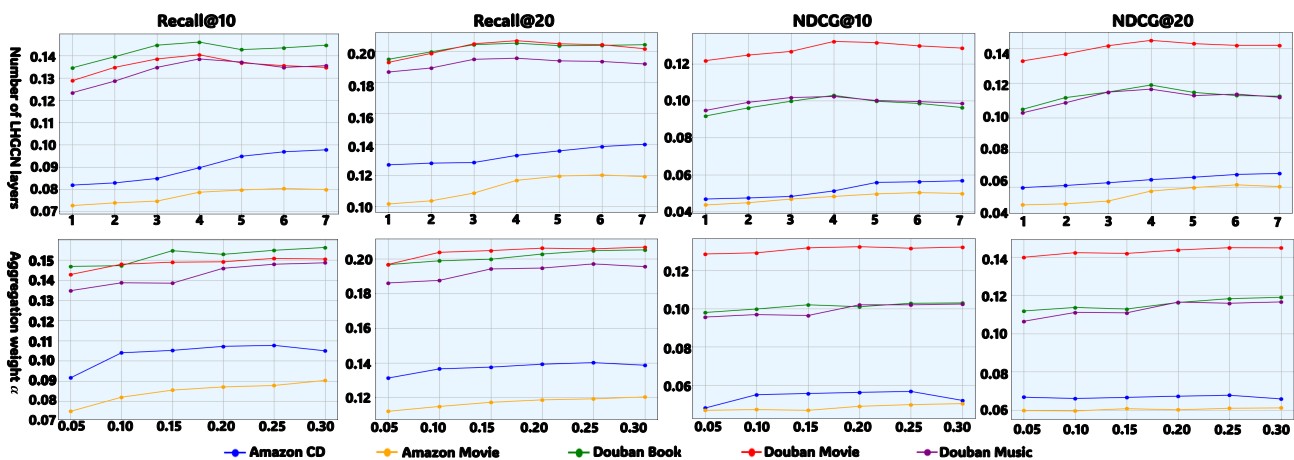

*Figure 7.* Sensitivity analysis on a number of LHGCN layers and aggregation weight $\alpha$

where $R_u$ is the set of items recommended to user $u$, $T$ is the set of tail items and $\sigma$ is an indicator function. This metric gives the proportion of head items and tail items among all the items recommended by the model. We conducted evaluations on the Amazon CD and Amazon Movie datasets. As shown in the upper part of Fig. 6, it can be observed that the Euclidean space-based model (LightGCN) recommends almost only head items to users in both datasets (97.8% in the Amazon CD dataset & 95.1% in the Amazon Movie dataset, 99.3% in Douban Movie dataset and 99.5% in Douban Music dataset). This indicates Euclidean space-based models tend to overlook tail items in CF tasks and aggravate the Matthew Effect. Conversely, this phenomenon is significantly mitigated by the three hyperbolic space-based models, showing that hyperbolic space is more suitable for the long-tail setting in CF. It can be observed that the model's emphasis on tail items differs between the Amazon and Douban datasets, which is primarily due to their significant differences in the density and data distribution.

### E.1.2. ANALYSIS OF MODEL'S PERFORMANCE ON TAIL ITEMS

To further investigate the performance of each hyperbolic-based model on tail items, in the second experiment, we evaluate the models' performance solely on tail items by calculating the recall@10 metric. As shown in Fig. 6, Hgformer significantly outperforms the other two hyperbolic space-based models on tail items. This is because Hgformer not only leverages the hyperbolic manifold to capture the hierarchical structure of the data but also introduces hyperbolic cross-attention which is beneficial for capturing global information. This allows the model to gather more information for tail nodes during the message-passing process, thereby improving the accuracy of recommendation.

### E.2. Sensitivity analysis(RQ5)

To evaluate the stability of our model, we conducted a sensitivity analysis on four hyperparameters of our model: number of LHGCN layers and aggregation weight $\alpha$. The results are shown in Fig. 7.

1. Our model remains relatively stable within a certain range for both aggregation weight and the number of LHGCN layers.
2. In the sensitivity analysis of aggregation weights, our model demonstrated relative stability in the range of 0.2 to 0.3, achieving better performance within this interval.
3. For the number of LHGCN layers, we observe that the optimal performance range of the model differs between the Amazon dataset and the Douban dataset. For the Amazon dataset, the optimal number of LHGCN layers falls within the range of 6 to 7, whereas for the Douban dataset, the optimal performance is achieved roughly between 4 and 5 layers.

$$\text{HR@K} = \frac{1}{|U|} \sum_{u \in U} \delta\big( \widehat{R}(u) \cap R(u) \neq \emptyset \big)$$

