# OpenReview forum: "Hgformer: Hyperbolic Graph Transformer for Collaborative Filtering"
_ICML.cc/2025/Conference — ICML 2025 poster_

### Official Review · Reviewer_A1RV · 2025-03-10

**Overall Recommendation:** 3

**Summary:**

The authors propose Hgformer: Hyperbolic Graph Transformer for Collaborative Filtering	and conduct extensive experiments to	analyze the proposed method.

**Claims And Evidence:**

Yes.

**Essential References Not Discussed:**

No.

**Experimental Designs Or Analyses:**

Yes.

**Methods And Evaluation Criteria:**

Yes.

**Other Comments Or Suggestions:**

See	“Other	Strengths	And	Weaknesses”

**Other Strengths And Weaknesses:**

Strengths

1.	The	paper is well-structured.

2.	The	experiments are comprehensive.

Weaknesses

1. The two key problems addressed in the paper, local structure modeling and embedding	distortion, have already	been extensively studied in previous works.

2. The combination of hyperbolic space and transformers has also	been explored	before. For	instance,	see: Hypformer: Exploring efficient transformer fully in hyperbolic space, KDD 2024.

3. Using hyperbolic space introduces	additional computational overhead. Is	this trade-off	justified?	 Moreover, how	does	 the	computational	cost	of the proposed	method	compare	with	existing	approaches?

4. The hyperparameter settings	 for	the baseline models are	not	specified	in the paper.

**Questions For Authors:**

None

**Relation To Broader Scientific Literature:**

Good.

**Theoretical Claims:**

Yes.

---

> ### Author Rebuttal · Authors · 2025-03-31
>
> **Q: The two key problems addressed in the paper have already been extensively studied in previous works.**
>
> **A:** Thank you for pointing out these issues. As you mentioned, these two problems have indeed been discussed in prior works, and we have acknowledged these discussions in our Introduction section. However, as we emphasize, these challenges remain unresolved in the recommendation domain. Specifically, as demonstrated in Section E.1 of the Appendix, our proposed method addresses long-tail problems better than previous approaches. Moreover, simultaneously tackling both issues is non-trivial, and as highlighted in the Challenges parts of our Introduction, there are still significant difficulties that need to be overcome. Additionally, extending kernel functions to hyperbolic space to enable the application of hyperbolic transformers in recommendation is a novel approach.
>
> **Q: The combination of hyperbolic space and transformers has also been explored.**
>
> **A:** Thank you for pointing out the issue. We have also noted the paper "Hypformer: Exploring Efficient Transformer Fully in Hyperbolic Space" and discussed the key difference in the Related Works section, which can be summarized as:
> 1) They focus on node classification, whereas our Hgformer is specifically tailored for collaborative filtering.
> 2) We propose LHGCN for effective local structure modeling in hyperbolic space, which is not explored in their work.
> 3) To achieve linear computational complexity, Hypformer first swaps the multiplication order of space-like values in self-attention vectors and then recalibrates the time-like values. In contrast, we adopt the cross-attention for better performance in CF tasks and directly extends the kernel function to hyperbolic space, which has a theoretical guarantee in approximation errors.
>
> Overall, we are the first to propose a hyperbolic graph transformer in the context of collaborative filtering.
>
> **Q: Using hyperbolic space introduces additional computational overhead. Is this trade-off justified? How does the computational cost of the proposed method compare with existing approaches?**
>
> **A:** Thanks for your question. The operations in hyperbolic space do not differ significantly from those in Euclidean space in terms of computational complexity.  The main parts of our model include LHGCN and Hyperbolic Graph Transformer. For LHGCN parts, In Section 2.2, we define the convolution in hyperbolic space by
>
> $$
> \mathbf{u}_i^{(l+1)} = \mathbf{Centroid}\bigl(\{\mathbf{u}_i^{(l)}, \{\mathbf{i}_k^{(l)} : k \in N_i\}\}\bigr)
> $$
>  and
>  $$ \mathbf{i}_j^{(l+1)} = \mathbf{Centroid}\bigl(\{\mathbf{i}_j^{(l)}, \{\mathbf{u}_k^{(l)} : k \in N_j\}\}\bigr).$$
>
> According to Definition B.7 in Appendix B, the above Centroid formula still yields an overall complexity  $O(|E| \times d)$ for an entire pass over the graph, where $|E|$ is the number of edges and $d$ is the embedding dimension and we find that the computational complexity is similar to that of the LightGCN convolution process.
>
> The complexity of the Hyperbolic Graph Transformer is detailed in Figure 3 in our paper and we can find that it is consistent with that of the NodeFormer, which is a graph transformer model in Euclidean space and is capable of dealing large-scale graphs. Hence, the hyperbolic variant retains the same order of computational complexity as its Euclidean counterpart.
>
> **Q: The hyperparameter settings for the baseline models are not specified in the paper.**
>
> **A:** Thanks for pointing out the problem. Below we provide the hyperparameter tuning ranges for baselines.
>
> | **Model**    | **Hyperparameters (Ranges)**                                              |
> |--------------|---------------------------------------------------------------------------|
> | **BPR**      | learning_rate: (2e-3, 1e-3, 5e-4)                                         |
> | **NGCF**     | learning_rate: (2e-3, 1e-3, 5e-4); node_dropout: (0.0, 0.1); message_dropout: (0.0, 0.1) |
> | **LightGCN** | learning_rate: (2e-3, 1e-3, 5e-4); n_layers: (2, 3); reg_weight: (1e-4, 1e-5) |
> | **HMLET**    | learning_rate: (2e-3, 1e-3, 5e-4); n_layers: (3, 4); activation_function: (elu, leakyrelu) |
> | **HGCF**     | learning_rate: (2e-3, 1e-3, 5e-4); n_layers: (2–5); scale: 0.1; learner: (adam, rsgd); margin: (0.1, 0.15, 0.2) |
> | **HICF**     | learning_rate: (2e-3, 1e-3, 5e-4); n_layers: (2–5); scale: 0.1; learner: (adam, rsgd); margin: (0.1, 0.15, 0.2) |
> | **HRCF**     | learning_rate: (2e-3, 1e-3, 5e-4); n_layers: (3–7); scale: 0.1; learner: (adam, rsgd); margin: (0.1, 0.15, 0.2) |
> | **SGFormer** | learning_rate: (2e-3, 1e-3, 5e-4); n_layers: 1; weight_decay: (1e-5, 1e-4, 1e-3); dropout_ratio: (0.0, 0.2, 0.3); num_heads: (2, 3) |
> | **NodeFormer** | learning_rate: (2e-3, 1e-3, 5e-4); n_layers: (2, 3); rb_order: (1, 2); num_heads: (2, 3)                                     |
> | **Hypformer**  |  learning_rate: (2e-3, 1e-3, 5e-4); n_layers: 1; num_heads: (2, 3); margin: (0.1, 0.15, 0.2)

---

### Official Review · Reviewer_n8SP · 2025-03-13

**Overall Recommendation:** 4

**Summary:**

The paper introduces Hgformer, a novel Hyperbolic Graph Transformer framework designed to address two critical limitations in GNN-based collaborative filtering (CF): local structure bias caused by neighborhood aggregation and embedding distortion in Euclidean space. The proposed method combines a parameter-free Local Hyperbolic Graph Convolutional Network (LHGCN), which performs graph convolution entirely within the hyperbolic manifold to avoid information loss from tangent space projections, with a Hyperbolic Cross-Attention mechanism that captures global user-item interactions in bipartite graphs. To ensure scalability, the authors propose a linear-complexity approximation for the cross-attention, supported by theoretical guarantees of unbiased estimation and controllable error bounds. The numerical experiments show that Hgformer is superior to leading CF models. Moreover, compared with traditional hyperbolic graph neural network methods, this approach can further enhance the model’s performance on long-tail items. Overall, the work presents a compelling integration of hyperbolic geometry and transformers, advancing CF research with both theoretical and practical contributions.

**Claims And Evidence:**

Claim 1: LHGCN outperforms tangent-space projection-based methods (e.g., HGCF).
Evidence: Figure 2(b) illustrates LHGCN’s direct hyperbolic aggregation vs. HGCF’s tangent-space mapping. Table 1 and ablation studies (Figure 4) show LHGCN achieves higher Recall/NDCG on most datasets.
Claim 2: Hyperbolic cross-attention effectively captures global user-item interactions. Evidence: Removing the transformer module causes significant performance drops (e.g., -11.6% Recall@10 on Amazon Book). Case studies (Figure 5) highlight its ability to recommend distant tail items. Additionally, tail-item analysis (Appendix E) shows that Hgformer improves the recommendation of long-tail items compared to models without hyperbolic cross-attention.
Claim 3: The linear-complexity approximation maintains performance while reducing costs.
Evidence: Theorems 3.1–3.2 provide theoretical guarantees. Figure 3 visualizes the approximation workflow. However, empirical validation of training/inference time on large-scale data is missing.

**Essential References Not Discussed:**

No.

**Experimental Designs Or Analyses:**

1.Comprehensive experiments on 5 datasets with ablation studies and case analyses.
2.Code availability enhances reproducibility.

**Methods And Evaluation Criteria:**

Method Soundness:
1.LHGCN’s design aligns with LightGCN’s simplicity, leveraging hyperbolic centroids for neighbor aggregation.
2.The unbiased estimation approach for cross-attention is theoretically grounded but may require tuning hyperparameters (e.g., random feature dimension m).
Evaluation Adequacy:
1.Metrics (Recall@K, NDCG@K) are standard for CF.
2.Baselines include hyperbolic GNNs (HGCF, HICF) and graph transformers (SGFormer), but some (e.g., SGFormer) are not CF-specific, potentially affecting fairness.

**Other Comments Or Suggestions:**

No.

**Other Strengths And Weaknesses:**

Strengths:
Novel integration of hyperbolic geometry and transformers for CF.
Strong empirical results and theoretical contributions.
Weaknesses:
Limited comparison with 2023–2024 hyperbolic CF models.
There is inconsistent formula formatting (e.g., lines 303–307 and Equation 12).

**Questions For Authors:**

1.	Limited comparison with 2023–2024 hyperbolic CF models. How does Hgformer compare to recent hyperbolic CF methods in terms of performance and scalability?
2.	Has a hyperparameter sensitivity analysis been conducted (e.g., curvature K, m)?
3.	Can you clarify the negative sampling strategies used for the margin-ranking loss?
4.	There is inconsistent formula formatting (e.g., Equation 12 and lines 304–307).

**Relation To Broader Scientific Literature:**

The paper connects hyperbolic GNNs (e.g., HGCF, HICF) and graph transformers, advancing CF through hyperbolic-local-global fusion.

**Theoretical Claims:**

Theorems 3.1 (unbiased estimation) and 3.2 (error bound) in Appendix C are logically sound.

---

> ### Author Rebuttal · Authors · 2025-03-31
>
> **Q: Empirical validation of training/inference time on large-scale data is missing.**
>
> **A:** Thank you for pointing this out. We have conducted empirical evaluations and reported the average computational time per epoch for several representative baselines and our proposed model on Amazon Book (our largest dataset) and Amazon Movie. In each cell, the first value represents the training time per epoch, while the second value represents the testing time.
> | Dataset       | BPR         | LightGCN    | HMLET       | HGCF        | Hgformer    |
> |---------------|-------------|-------------|-------------|-------------|-------------|
> | Amazon Book   | 0.46s/0.74s | 2.22s/1.33s | 12.63s/1.28s| 6.06s/1.84s | 14.93s/1.92s|
> | Amazon Movie  | 0.22s/0.56s | 1.41s/0.71s | 7.41s/0.5s  | 3.03s/0.78s | 9.8s/0.83s  |
>
> **Q: Limited comparison with 2023–2024 hyperbolic CF models. How does Hgformer compare to recent hyperbolic CF methods in terms of performance and scalability?**
>
> **A:** Thank you for pointing out the issue. We have looked into recent literature from 2023–2024 and plan to include HyperCL[1], which extend contrastive learning into hyperbolic space and applied it for collaborative filtering, as one of our baselines. The results are as follows:
>
> | Dataset         | Metric                   | Our Method | HyperCL  |
> |----------------|--------------------------|------------|----------|
> | **Amazon Book** | Recall@10                | 0.9010      | 0.0756   |
> |                | Recall@20                | 0.1291     | 0.1189   |
> |                | NDCG@10                  | 0.0573     | 0.0482   |
> |                | NDCG@20                  | 0.0681     | 0.0582   |
> | **Amazon CD**   | Recall@10                | 0.0977     | 0.0828   |
> |                | Recall@20                | 0.1401     | 0.1221   |
> |                | NDCG@10                  | 0.0567     | 0.0496   |
> |                | NDCG@20                  | 0.0678     | 0.0598   |
> | **Amazon Movie**| Recall@10                | 0.0803     | 0.0776   |
> |                | Recall@20                | 0.1203     | 0.1186   |
> |                | NDCG@10                  | 0.0503     | 0.0492   |
> |                | NDCG@20                  | 0.0612     | 0.0594   |
> | **Douban Book** | Recall@10                | 0.1462     | 0.1392   |
> |                | Recall@20                | 0.2052     | 0.1968   |
> |                | NDCG@10                  | 0.1030      | 0.0976   |
> |                | NDCG@20                  | 0.1189     | 0.1126   |
> | **Douban Movie**| Recall@10                | 0.1405     | 0.1386   |
> |                | Recall@20                | 0.2068     | 0.2046   |
> |                | NDCG@10                  | 0.1322     | 0.1316   |
> |                | NDCG@20                  | 0.1447     | 0.1435   |
> | **Douban Music**| Recall@10                | 0.1386     | 0.1268   |
> |                | Recall@20                | 0.1955     | 0.1826   |
> |                | NDCG@10                  | 0.1024     | 0.0937   |
> |                | NDCG@20                  | 0.1165     | 0.1076   |
>
>
> **Q: Has a hyperparameter sensitivity analysis been conducted (e.g., curvature K, m)?**
>
> **A:** Thanks for the question. We provided a sensitivity analysis in Appendix E.2. However, we did not conduct a sensitivity analysis for K and m. We have now supplemented our work with an analysis for these two hyperparameters, and the results can be found at the link below. https://anonymous.4open.science/r/Hgformer-2AE0/sensitivity_analysis_rebuttal.pdf
>
> **Q:Can you clarify the negative sampling strategies used for the margin-ranking loss?**
>
> **A:** We adopted the default approach provided by the RecBole framework. Specifically, negative samples were uniformly drawn from items with which the user had no previous interaction. To ensure a fair and consistent comparison across different methods, all baselines utilizing negative sampling followed this same strategy and pair each positive sample with exactly one negative sample.
>
>
> **Q:There is inconsistent formula formatting (e.g., Equation 12 and lines 304–307).**
>
> **A:** Thanks for pointing our this inconsistency! We acknowledge that they can indeed be easily misunderstood. In lines 304–307, the $u^{final}$ and $i^{final}$  are derived from Equation 14, where $u^{global}$ and $i^{global}$ are obtained from Equation 5. Meanwhile, Equation 12 is introduced to reduce the computational complexity of Equation 3 (which leads to Equation 5) to linear complexity. We will add further clarifications in the paper to improve its readability.
>
> **References:**
>
> [1] Qin Z, Cheng W, Ding W, et al. Hyperbolic Graph Contrastive Learning for Collaborative Filtering[J]. IEEE Transactions on Knowledge and Data Engineering, 2024.

---

### Official Review · Reviewer_UU62 · 2025-03-14

**Overall Recommendation:** 2

**Summary:**

The paper proposes a Hyperbolic Graph Transformer architecture, to tackle the long-tail problems in CF tasks, which leverages LHGCN for graph structure modeling and hyperbolic cross-attention for global information modeling

**Claims And Evidence:**

Are the claims made in the submission supported by clear and convincing evidence?

**Essential References Not Discussed:**

The paper proposes Hyperbolic Transformer but does not cite "Hyperbolic Attention Networks" (Gulcehre et al., ICLR 2019), which specifically established the theoretical foundations for attention mechanisms in hyperbolic space.

the paper overlooks "Performers: Rethinking Attention with Performers" which introduced a kernel-based approximation method achieving linear complexity in Euclidean space using random feature maps.

Similarly, "Nyströmformer" is not discussed despite offering another established approach to linear-complexity attention.

For their long-tail recommendation contribution, the paper fails to cite "Causal Intervention for Leveraging Popularity Bias in Recommendation", which specifically addressed popularity bias with a causal inference approach.

**Experimental Designs Or Analyses:**

is there any validation set for these experiments? How does the author get these results?

there are no statistical significance tests (t-tests, confidence intervals) to verify if these differences are statistically meaningful or potentially due to random variation.

missing Euclidean space transformer-based recommenders like SASRec and BERT4Rec

regarding the baselines, SGFormer, NodeFormer, and Hypformer were not originally designed for recommendation tasks. It remains unclear how the authors adapted these models for recommendation tasks and comparison purposes.

**Methods And Evaluation Criteria:**

The author mentions not using tangent space for aggregation, but in the "Embedding Aggregation and Optimization" section, they still use aggregation, which seems inconsistent and unreasonable.

Regarding the baselines, SGFormer, NodeFormer, and Hypformer were not originally designed for recommendation tasks. It remains unclear how the authors adapted these models for recommendation tasks and comparison purposes.

**Other Comments Or Suggestions:**

None

**Other Strengths And Weaknesses:**

None

**Questions For Authors:**

None

**Relation To Broader Scientific Literature:**

The linear-complexity approximation builds upon the theoretical framework of Linformer but adapts it for hyperbolic manifolds—a mathematical contribution extending beyond recommendations.

**Theoretical Claims:**

While the paper claims LHGCN performs "graph convolution entirely in hyperbolic manifold without mapping back to Euclidean space" (p.2), the mathematical justification for why this preserves information better than alternative approaches is more empirically than theoretically supported.

---

> ### Author Rebuttal · Authors · 2025-03-31
>
> **Q: Inconsistency between two aggregation approaches**
>
> **A:** Sorry for the misunderstanding. These two types of aggregation serve different purposes in our model. The aggregation in Section 2.2 indicates gathering neighbor information for each node to capture multi-hop relationships, which is performed in multiple layers—thus, any distortion at this stage would accumulate significantly. In contrast, the aggregation of local and global information in  Section 2.4 is simply fusing information from the two representation space for each node, which is a much simpler operation and involves negligible information distortion. In fact, aggregation step in Section 2.4 can be easily replaced by the weighted centroid of two embeddings defined in Definition B4 in the Appendix, and we conducted additional comparative experiments:
> | **Dataset**       | **Metric**  | **Our method** | **Hyperbolic aggregation** |
> |-------------------|-------------|----------------|----------------------------|
> | **Amazon CD**     | Recall@10 | 0.0977         | 0.0978                     |
> |                   | Recall@20 | 0.1401         | 0.1398                     |
> |                   | NDCG@10   | 0.0567         | 0.0572                     |
> |                   | NDCG@20   | 0.0678         | 0.0680                     |
> | **Douban Book**   | Recall@10 | 0.1462         | 0.1459                     |
> |                   | Recall@20 | 0.2052         | 0.2045                     |
> |                   | NDCG@10   | 0.1030          | 0.0995                     |
> |                   | NDCG@20   | 0.1189         | 0.1186                     |
>
> We observe that the differences between the two methods are marginal.
>
> **Q: Implementations of SGFormer, NodeFormer and Hypformer**
>
> **A:** We adopted the encoder parts of these models straightforwardly and replace their output layers with recommendation-specific modules (For NodeFormer, SGFormer we applied BPR loss, which is the same as LightGCN and NGCF and for Hypformer, we applied Hyperbolic Margin Ranking Loss, which is the same as our hyperbolic baselines). The detailed implementations are publicly available in the anonymized GitHub repository in the Abstract.
>
> **Q: Lack of mathematical justification for LHGCN**
>
> **A:** Thanks for pointing out this issue. In the following, we provide a theoretical analysis supporting our claims.
>
> For a set of hyperbolic points $N \subset H^{d+1, K}$, for each point $x_i \in N$, the centroid is defined as the point $c \in H^{d+1, K}$ that minimizes the sum of squared hyperbolic distances:
>
> $Centroid(N) = \arg\min_{c \in H^{d+1, K}} \sum_{x_i \in N} (d^K(c,x_i))^2 = \arg\min_{c \in H^{d+1, K}} f(c).$
>
> Its closed-form solution is given by
>
> $c^* = \sqrt{K} \frac{\sum_{x_i \in N} x_i}{| \sum_{x_i \in N} x_i |_M}.$
>
> Another way is to aggregate at a north pole point $o$
>
> $\bar{c}' = \frac{1}{|N|}\sum_{x_i \in N} log_o(x_i),$
>
> and then map them back:
>
> $\bar{c}^* = exp_o(\bar{c}') = cosh\Bigl(\frac{|\bar{c}'|_M}{\sqrt{K}}\Bigr)o + \sqrt{K},sinh\Bigl(\frac{|\bar{c}'|_M}{\sqrt{K}}\Bigr)\frac{\bar{c}'}{|\bar{c}'|_M}.$
>
> Since $f(c)$ is convex in hyperbolic space (see [1]), it follows that
>
> $f\left(c^*\right) \le f\left(\bar{c}^*\right)$
>
> This means that aggregation in Euclidean space is only a suboptimal solution and will lead to distortion.
>
> **Q: Is there validation set? How do authors get the results?**
>
> **A:** Thanks for the questions. To ensure accuracy and fairness in our experiments, all the models were tested on RecBole, a widely accepted framework in the field of recommender systems. The dataset was split into training, validation, and test sets with an 8:1:1 ratio. During training, we adopted an early-stop mechanism for all models, stopping training if no improvement was observed on the validation set for 30 epochs and the best-performing parameters on the validation set were selected for final evaluation on the test set.
>
> **Q: No statistical significance tests (t-tests, confidence intervals)**
>
> **A:** We respectfully clarify that we calculate p-values for our experiments, which is depicted in the caption of table 1. Here, we provide the specific p-values and t-values:
> | Dataset       | p-value | t-value |
> |---------------|---------|---------|
> | Amazon Book   | 0.023   | 2.5     |
> | Amazon CD     | 0.019   | 2.6     |
> | Amazon Movie  | 0.021   | 2.7     |
> | Douban Book   | 0.018   | 2.9     |
> | Douban Movie  | 0.027   | 2.4     |
> | Douban Music  | 0.038   | 2.3     |
>
>  **Q: Essential References Not Discussed**
>
> **A:** Thanks for pointing out the problem. For "Hyperbolic Attention Networks", "Performers" and "Nyströmformer", we would like to add them into Graph Transformer part in Related Works and for "Causal Intervention for Leveraging Popularity Bias in Recommendation", "SASRec" and "BERT4Rec", we would like to add an extra section in Appendix for discussion.
>
>  **Reference:**
>
> [1]Bacák M. Computing medians and means in Hadamard spaces.

---

### Decision · Program_Chairs · 2025-05-01

**Decision:**

Accept (poster)

**Comment:**

Post-rebuttal, reviewers have diverse evaluation scores (weak reject, weak accept, accept). Based on such scores, I recommend acceptance. Please ensure that the suggested revisions are incorporated into the final version, particularly raised by Reviewer UU62:
- Limited novelty: The use of the centroid as an average in Lorentz GCN appears to overlap with previous methods such as LGCN and related frameworks.
- Statement-implementation inconsistency: Although the paper discusses issues with the tangent space, the method still relies on tangent space operations when integrating global and local information.
- Unfair comparison: The implementation of baseline methods, as described by the authors, may not faithfully reflect their intended usage, potentially affecting the fairness of the comparisons.